# Genotyping cognate *Plasmodium falciparum* in humans and mosquitoes to estimate onward transmission of asymptomatic infections

Kelsey M. Sumner [1,2], Elizabeth Freedman[2], Lucy Abel[3], Andrew Obala[4], Brian W. Pence[1], Amy Wesolowski [5], Steven R. Meshnick[1], Wendy Prudhomme-O'Meara[2,6,7,8] & Steve M. Taylor [2,7,8 ✉]

Malaria control may be enhanced by targeting reservoirs of *Plasmodium falciparum* transmission. One putative reservoir is asymptomatic malaria infections and the scale of their contribution to transmission in natural settings is not known. We assess the contribution of asymptomatic malaria to onward transmission using a 14-month longitudinal cohort of 239 participants in a high transmission site in Western Kenya. We identify *P. falciparum* in asymptomatically- and symptomatically-infected participants and naturally-fed mosquitoes from their households, genotype all parasites using deep sequencing of the parasite genes *pfama1* and *pfcsp*, and use haplotypes to infer participant-to-mosquito transmission through a probabilistic model. In 1,242 infections (1,039 in people and 203 in mosquitoes), we observe 229 (*pfcsp*) and 348 (*pfama1*) unique parasite haplotypes. Using these to link human and mosquito infections, compared with symptomatic infections, asymptomatic infections more than double the odds of transmission to a mosquito among people with both infection types (Odds Ratio: 2.56; 95% Confidence Interval (CI): 1.36–4.81) and among all participants (OR 2.66; 95% CI: 2.05–3.47). Overall, 94.6% (95% CI: 93.1–95.8%) of mosquito infections likely resulted from asymptomatic infections. In high transmission areas, asymptomatic infections are the major contributor to mosquito infections and may be targeted as a component of transmission reduction.

[1] Department of Epidemiology, Gillings School of Global Public Health, University of North Carolina, Chapel Hill, NC, USA. [2] Division of Infectious Diseases, School of Medicine, Duke University, Durham, NC, USA. [3] Academic Model Providing Access to Healthcare, Moi Teaching and Referral Hospital, Eldoret, Kenya. [4] School of Medicine, College of Health Sciences, Moi University, Eldoret, Kenya. [5] Department of Epidemiology, Johns Hopkins Bloomberg School of Public Health, Baltimore, MD, USA. [6] School of Public Health, College of Health Sciences, Moi University, Eldoret, Kenya. [7] Duke Global Health Institute, Duke University, Durham, NC, USA. [8] These authors contributed equally: Wendy Prudhomme-O'Meara, Steve M. Taylor. ✉email: steve.taylor@duke.edu

Despite sustained malaria prevention efforts, progress in malaria control has stalled since 2010, with 228 million malaria episodes in 2018[1]. This persistence could result from a failure to target and mitigate infections in individuals or populations that disproportionally contribute to malaria transmission, so called malaria reservoirs. Sustained *Plasmodium falciparum* transmission despite case reductions could result from asymptomatic *P. falciparum* infections[2–7]. Asymptomatic infections are defined as the presence of parasites in the blood at any density in the absence of malaria-like symptoms[8] and typically represent either a state prior to development of symptoms (i.e., pre-symptomatic)[8] or one in which symptoms are attenuated due to non-sterilizing adaptive immunity[9]. Asymptomatic infections include both submicroscopic and microscopically patent infections that have different capacities for infecting mosquitoes[10]. Because asymptomatic infections are sub-clinical and therefore often remain untreated[11], asymptomatically infected people can remain infectious to mosquitoes for prolonged periods and fuel onward transmission despite control measures[12,13].

The relative contribution of asymptomatic infections to overall malaria transmission is incompletely understood. Several studies have compared the transmission potential of asymptomatic and symptomatic *P. falciparum* infections to mosquitoes and have generally confirmed that such infections are transmissible[14–18]. However, the small sample sizes and the use of experimental approaches using artificial membrane feeding by laboratory-reared mosquitoes limit generalizability by failing to capture variations in human activity, vector complexity and behavior, and parasite biology that influence transmissibility in natural settings. Such controlled feeding studies are critical to understand the fundamental biology of parasite transmission, and studies in natural, uncontrolled settings are necessary to confidently extend these insights to understand how they shape disease epidemiology. It is particularly critical to understand the impact of these infections in complex high-transmission settings, in which asymptomatic infections are highly prevalent but not commonly prioritized in transmission-reduction efforts. Such efforts include enhanced testing, treatment, and prevention on either mass or focal scales, and these tools can be employed more efficiently and rationally with a better understanding of the relative transmissibility of asymptomatic *P. falciparum* infections.

We investigated the contribution of asymptomatic *P. falciparum* infections to successful mosquito infection in a 14-month longitudinal cohort of 239 people in Western Kenya, a hyperendemic area where asymptomatic infections are common[19,20]. In these households, we collected cognate infections in both people and indoor-resting Anopheline mosquitoes under the premise that, owing to the endophilic and endophagic preferences for feeding by the principal vectors *Anopheles gambiae* and *Anopheles funestus*, household transmission would be both measurable and substantial. Building upon previous studies, our approach combines empirical data collection of naturally fed mosquitoes, parasite genotyping using amplicon deep sequencing, and probabilistic modeling to estimate the transmissibility from people to mosquitoes of asymptomatic relative to symptomatic *P. falciparum* infections. We hypothesized that, compared to symptomatic infections, asymptomatic infections would be a larger source of infected mosquitoes.

## Results

From June 2017, we enrolled 268 participants across 3 villages in Bungoma county, Kenya in the cohort study; after excluding participants with either zero *P. falciparum* infections or <2 months of follow-up, the analysis data set consisted of 239 participants across 38 households who were visited monthly for active case detection of asymptomatic infections and as needed for passive case detection of symptomatic infections. In these participants across 14 months, we recorded 137 symptomatic *P. falciparum* infections during 501 sick visits and 902 asymptomatic *P. falciparum* infections during 2312 routine visits (Fig. 1). From their households, we collected 1494 female *Anopheles* mosquitoes; of the 1450 mosquito abdomens with genomic DNA (gDNA) available, we identified 203 *P. falciparum*-positive mosquitoes.

These 1242 real-time PCR-positive *P. falciparum* infections (N = 902 asymptomatic infections, N = 137 symptomatic infections, and N = 203 infected mosquito abdomens) were genotyped for the *P. falciparum* parasite genes encoding apical membrane antigen-1 (*pfama1*) and circumsporozoite protein (*pfcsp*) using PCR amplification, amplicon deep sequencing, and a validated haplotype inference program with strict quality-filtering criteria[21]. *Pfcsp* and *pfama1* were selected owing not to phenotypes associated with their protein products but rather to their sequence diversity, which enables capture of diverse parasite strains and matching strains between hosts[22]. Results for *pfama1* haplotypes are reported in the supplement. For *pfcsp*, we obtained analyzable haplotypes that passed our custom quality filtering for 1046 samples (84.2%), across which we identified 229 unique *pfcsp* haplotypes. These haplotypes harbored variants at 72 nucleotide positions in the sequenced segment of *pfcsp*; variants at 37 (51.4%) of these positions were previously reported (Fig. S2)[23–25]. Many haplotypes were observed across all three sample types, but some haplotypes were private to asymptomatic infections, symptomatic infections, or mosquitoes (Figs. 2 and S12 and Table S6). Between sample types, the median *pfcsp* multiplicity of infection (MOI) was higher for mosquitoes (6, interquartile range [IQR]: 4–9) compared to either symptomatic infections (1, IQR: 1–3, *p* value <0.001 by pairwise Wilcoxon Rank-Sum test) or asymptomatic infections (3, IQR: 1–7, *p* value <0.001) (Fig. 2).

We used these *P. falciparum* haplotypes as identifiers by which to estimate parasite transmission from people to mosquitoes by computing pairwise metrics of parasite haplotype sharing between infected participants and mosquitoes. We first analyzed a subset of 65 participants who suffered both asymptomatic and symptomatic infections, from whom we paired all 225 infected events (N = 143 asymptomatic and N = 82 symptomatic infections) with infected mosquitoes that were collected: (i) between 7 days before and 14 days after the event, and (ii) within 3 km of the participant's household. This yielded 1565 participant–mosquito pairs for the 225 events; this subset of participants and events was similar to the overall population (Table S1). For each event, we computed the proportion of participant–mosquito pairings in which at least 1 *pfcsp* haplotype was shared between the mosquito and the participant. In a multi-level logistic regression model controlling for parasite density and mosquito abundance, compared to their symptomatic infections, their asymptomatic infections had higher odds of sharing a parasite haplotype with infected mosquitoes [odds ratio (OR): 2.56 (95% confidence interval (CI): 1.36–4.81)] (Fig. 3). Results were similar but not statistically significant in parallel analyses using *pfama1* [OR: 1.30 (95% CI: 0.63–2.69)] (Figs. S13 and S14), indicating that, compared to when the individuals suffered symptomatic infections, their asymptomatic infections were more likely to result in successful parasite transmission to mosquito vectors.

In order to more comprehensively analyze transmission across all participants irrespective of their infection counts, we extended our assessment of transmission using pairings of all infections in participants and mosquitoes using a probabilistic model of transmission. Across all samples over 14 months, there were 159,285 potential pairings of infected participants and

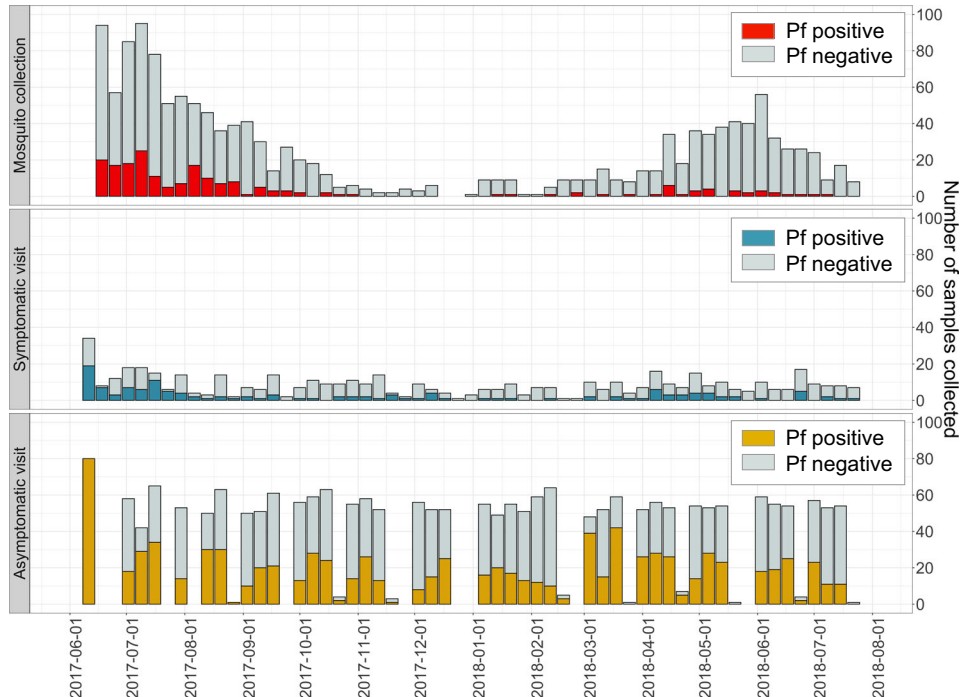

**Fig. 1 *P. falciparum* infections observed across study participants and female *Anopheles* mosquitoes across 14 months.** Female Anopheline mosquitoes were captured weekly by vacuum aspiration and their abdomens were tested using real-time PCR for the presence (red) or absence (gray) of *P. falciparum* (Pf) parasites. Symptomatic malaria infections were captured by passive case detection with clinical symptoms and positive *P. falciparum* results by both RDT and real-time PCR (blue). The number of participants who had malaria-like symptoms and requested a symptomatic visit but did not have a confirmed symptomatic infection were also identified (gray). Asymptomatic malaria infections were captured by active case detection at monthly follow-up visits with participants and real-time PCR-positive for *P. falciparum*. These monthly visits were conducted in different weeks for each of the three villages, with additional re-visits if needed to sample enrolled participants who were absent for the initial visit. Monthly counts of asymptomatic malaria infections (yellow) and uninfected participants (gray) were reported.

mosquitoes, and after applying the aforementioned temporal and geographic distance constraints to these pairings to remove those with implausible transmission potential (Fig. 4), the final analysis data set consisted of 3727 participant–mosquito pairs. These comprised 198 participants and 182 mosquitoes that were drawn from 37 households across all 3 villages. Among these 3727 pairings, mosquitoes paired with asymptomatic participants ($N = 3012$) outnumbered those paired with symptomatic participants ($N = 715$). Compared to those including asymptomatic infections, pairings including symptomatic infections had higher parasite densities ($p$ value <0.001 by Wilcoxon Rank-Sum test), were more likely to occur in a participant <5 or >15 years ($p$ value <0.001 by Pearson's $\chi^2$ test; Tables 1 and S2), and typically occurred during periods with larger mosquito abundance ($p$ value <0.001 by Pearson's $\chi^2$ test). Across the all pairings, the median number of haplotypes shared within a participant–mosquito pair was 1 (range: 0–8, IQR: 0–2) for asymptomatic and 0 (range: 0–7, IQR: 0–1) for symptomatic infections.

For each of these 3727 pairings, we computed the probability that a shared haplotype between a participant and mosquito represented a transmission event [$P(\text{TE}_\text{all})$] as a function of three indices: (i) temporal distance [$P(\text{TE}_\text{t})$], (ii) geographic distance [$P(\text{TE}_\text{d})$], and (iii) the prevalence and quantity of shared haplotypes between samples [$P(\text{TE}_\text{h})$] (Fig. 4). The rationale for this approach was to assign a probability to each pair that reflected the level of confidence that the pair represented a participant-to-mosquito transmission event. The probability increased for pairs that were closer in space or time and for those which shared a higher number of haplotypes or haplotypes that were comparatively rare across samples. We aggregated all three terms into a compound estimate of a probable transmission event [$P(\text{TE}_\text{all})$];

across all pairings ($N = 3727$), the median $P(\text{TE}_\text{all})$, was 0.05 (IQR: 0.00–0.15), and among only those pairings with at least 1 shared haplotype ($N = 2278$), the median $P(\text{TE}_\text{all})$ was 0.12 (IQR: 0.06–0.21). $P(\text{TE}_\text{all})$ represented a relative likelihood that a human and mosquito pair that shared parasite haplotypes represented a transmission event and should be interpreted relative to other $P(\text{TE}_\text{all})$ values.

We compared our estimates of transmission for each pairing between those with asymptomatic and symptomatic infections using a multi-level logistic regression on $P(\text{TE}_\text{all})$, controlling for parasite density, participant age, mosquito abundance, and village. Using *pfcsp* haplotypes for haplotype indices, compared to symptomatic infections, asymptomatic infections had 50% higher odds of being matched to a mosquito infection (OR: 1.50, 95% CI: 1.07–2.10; Fig. 5). In parallel analyses using *pfama1*, we observed a similar increase in the odds of transmission to mosquitoes from asymptomatic compared to symptomatic infections (OR: 1.22 95% CI: 0.82–1.82; Fig. S15). We re-computed regression models after dichotomizing our estimated $P(\text{TE}_\text{all})$ values at various cutoffs from 0.00 to 0.55, reflecting the range of $P(\text{TE}_\text{all})$ values and increasing stringency for defining a transmission event (Fig. 5). Across this broad range of $P(\text{TE}_\text{all})$ definitions, asymptomatic infections had consistently higher odds of onward parasite transmission. When we defined a transmission event as any non-zero $P(\text{TE}_\text{all})$ value, reflecting sharing between participant and mosquito of any number and quality of haplotypes, compared to symptomatic infections, asymptomatic infections more than doubled the odds of transmission to a mosquito (OR: 2.66, 95% CI: 2.05–3.47).

Finally, we used this measurement to estimate the contribution of each type of infection to onward transmission across our

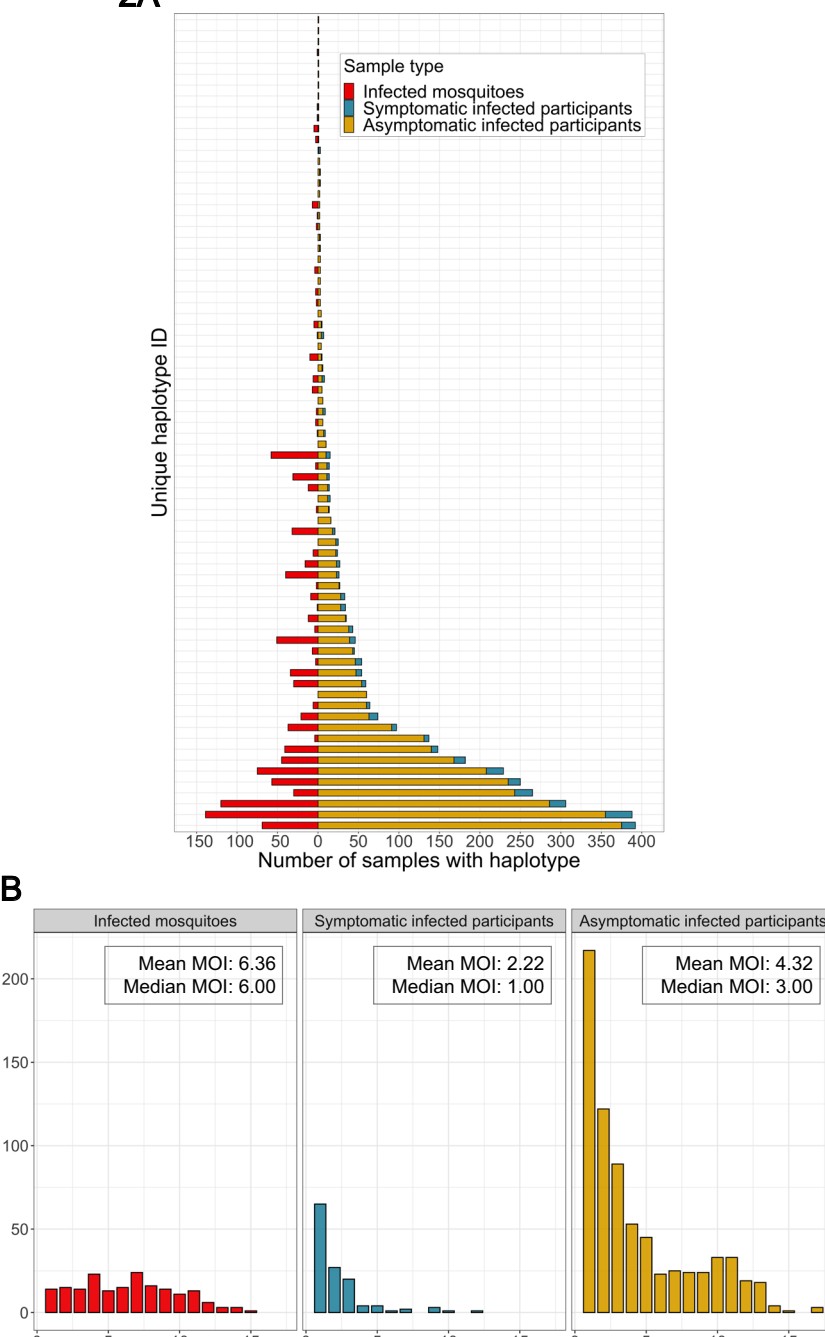

**Fig. 2 Distributions of *pfcsp* haplotypes across and within participants and mosquitoes. A** Distribution of the 75 most common *pfcsp* haplotypes in mosquitoes (red), symptomatic infections (blue), and asymptomatic infections (yellow), ordered vertically by the number in the asymptomatic infections. A full plot of all 229 *pfcsp* haplotypes and a table of the counts for these haplotypes across sample types is in the supplement (S12 Fig and Table S6). **B** Multiplicity of infection (MOI) based on observed number unique *pfcsp* haplotypes in each mosquito abdomen (red), symptomatic infection (blue), and asymptomatic infection (yellow).

population as a function of the monthly proportion of all infections that were asymptomatic, which varied from 73.4 to 97.4% between months. Using these, we estimated that monthly contributions to mosquito infections by asymptomatic infections varied from 88.0 to 99.0% (Fig. 5), and averaged across all months in our high and perennial transmission setting, asymptomatic infections were the source of 94.6% (95% CI: 93.1–95.8%) of mosquito infections.

## Discussion

In this longitudinal epidemiological and entomological cohort in Western Kenya, we investigated the relative contributions to onward *P. falciparum* transmission of asymptomatic compared to symptomatic *P. falciparum* infections. To do so, we analyzed parasite haplotypes in people and mosquitoes using a probabilistic model to directly estimate participant-to-mosquito malaria transmission. We report that, compared to symptomatic people,

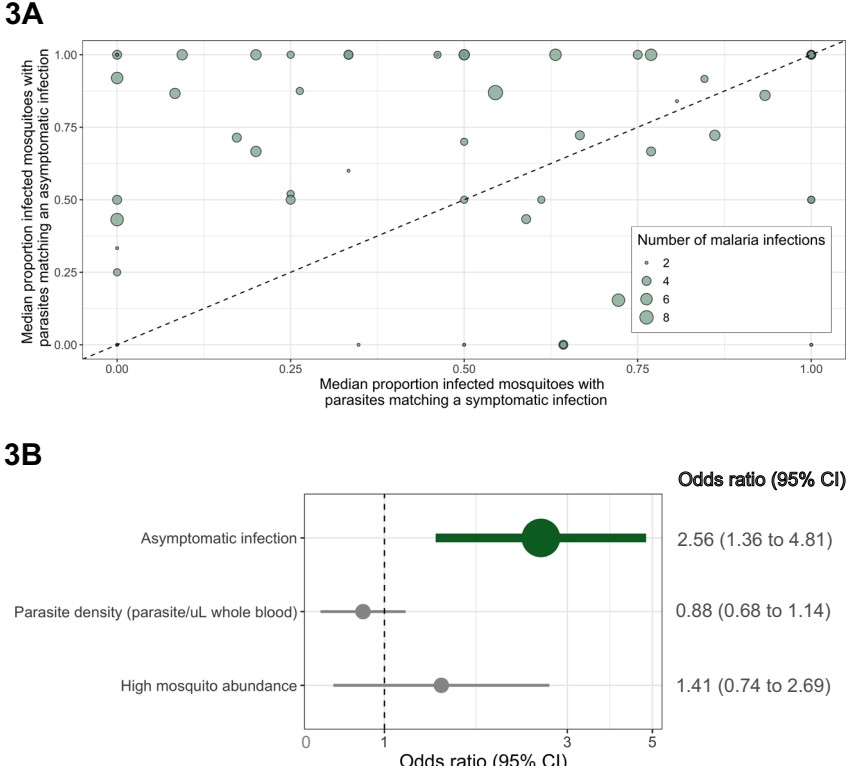

**Fig. 3 Comparison of the proportion of infected mosquitoes harboring a matching *pfcsp* haplotype for participants with both asymptomatic and symptomatic infections. A** Scatterplot of the proportion of pairings with a mosquito that shared a minimum of one haplotype for asymptomatic (*y*-axis) and symptomatic (*x*-axis) infections. Each dot is a participant who suffered at least one asymptomatic and symptomatic infection, and for participants with more than one of either type of infection, the plotted value is the median of proportions across infections within that type. Size of dots is relative to the total number of the participant's infections. **B** Odds ratios of the proportion of matched mosquitoes in a multi-level logistic regression model using the continuous coding of the proportion of participant–mosquito pairings that shared haplotypes for each infection (N = 1565 participant–mosquito pairings). Data are presented as odds ratios (dots) with 95% confidence intervals (bars).

those with asymptomatic infections had more than double the odds of transmission to mosquitoes. Owing to this as well as the high prevalence of asymptomatic infections, we estimated that asymptomatic infections were the source of nearly all *P. falciparum* parasites infecting mosquitoes. Our findings provide an explicit rationale to target asymptomatic *P. falciparum* infections as a component of transmission-reducing programs.

Across 14 months of observation in a high-transmission setting, asymptomatic *P. falciparum* infections were the major source of onward malaria transmission. Specifically, relative to symptomatic infections, asymptomatic infections had 2.66-fold odds of probable malaria transmission to mosquitoes. Our findings are consistent with results from smaller studies, which suggested that asymptomatic infections were more likely to transmit to mosquitoes than symptomatic infections[16,17]. Those studies used experimental feeding on infected blood by laboratory-reared mosquitoes to measure transmission and therefore could not capture variance in the feeding behaviors of vectors[26] or the natural trajectories of infections[27]. Our findings build upon these previous studies by capturing participant-to-mosquito transmission longitudinally in a larger study population and in a natural setting with mosquitoes collected within participants' households. Notably, we also observed this positive association between asymptomatic infections and transmission among the overall cohort using the alternate, unlinked parasite genotyping locus of *pfama1* as well as among a subset of participants who suffered both asymptomatic and symptomatic infections during the study period. Although our study does not enable the identification of a clear mechanism for this association, a lack of symptoms may

allow a longer duration of infection and thereby enable both the development of gametocytes as well as more opportunities to be bitten by and transmit malaria to mosquitoes[11].

Our approach used probabilistic modeling of genotypes captured by amplicon deep sequencing to estimate *P. falciparum* transmission. Prior studies of participant-to-mosquito malaria transmission using alternate approaches[14–18] have incompletely captured the complexity of natural systems, which limits their generalizability. Mosquito feeding experiments employing either direct skin or membrane feeding fail to represent numerous participant-, mosquito-, and parasite-related factors that are critical to transmission. These critical factors include variance among mosquito vectors in biting preferences[28], behaviors[26], and success[29]; among parasites in replication rates[27] and gametocyte production[30]; and among participants in exposure to vectors[31] and care-seeking behavior[32]. Similarly, this complexity also confounds the use of gametocyte prevalence or density as a proxy for transmission[33], which may more precisely define which infections can rather than do transmit. Other studies have used modeling approaches to estimate how transmission dynamics could change in a more realistic setting, finding that submicroscopic infections are a large source of malaria spread[34,35]; however, the studies did not examine how transmission differed by symptomaticity.

This approach to measure participant-to-mosquito transmission offers a scalable tool that can be adapted to diverse settings. Consistent with prior reports from high-transmission settings[22,36–39], we found high diversity of *pfama1* and *pfcsp* haplotypes in our study site, likely the result of strong balancing

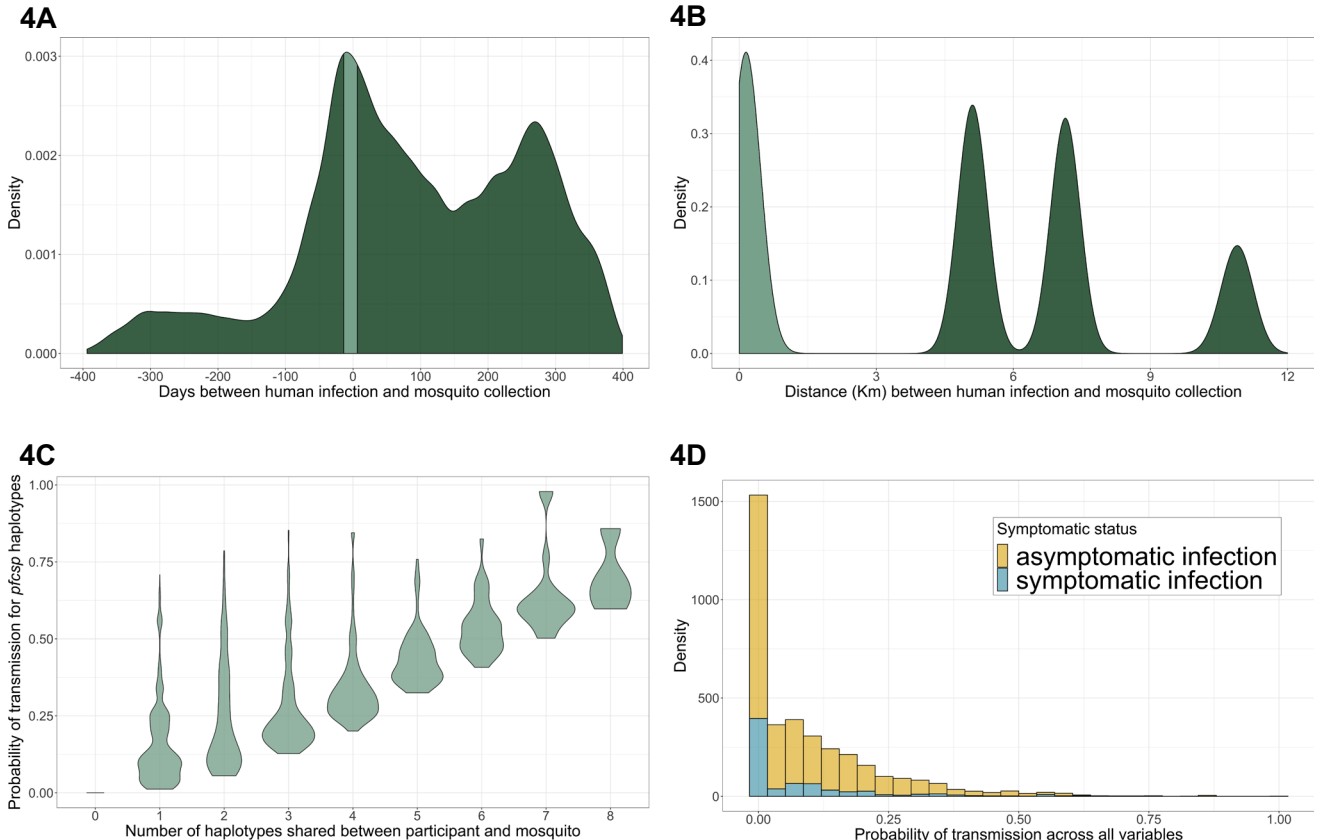

**Fig. 4 Modeling approach to estimate the probability of a *P. falciparum* transmission event to mosquitoes using the *pfcsp* gene target. A** Distribution of the interval in days between all possible pairings ($N = 159{,}285$) of all infected participants and mosquito abdomens. Day 0 was set as the date of the mosquito infection, and therefore negative values indicate that the mosquito was collected prior to the participant infection. The light green area indicates those pairings in which the mosquito was collected within 7 days prior to or 14 days after the participant's infection. Subsequent analysis was restricted to these pairings. **B** Distribution of the distance interval between all possible pairings of infected participants and mosquito abdomens. The light green area indicates those pairings within the same village and at a maximum distance of 3 km, to which subsequent analysis was restricted. Across these pairings, a probability function was applied (Fig. S7) to upweight pairings with shorter distance intervals. The peaks result from differences in distance across the three villages. **C** Distribution of the estimated probabilities of transmission as a function of the number of *pfcsp* haplotypes shared within the participant–mosquito pair. These probabilities were estimated by upweighting pairings which shared more haplotypes and which shared haplotypes that were rare across the entire study population. **D** Distributions of the final estimated probabilities of transmission events stratified by symptomatic status of the participant infection. Final probabilities were computed as the product of the individual probabilities based on the time interval, distance interval, and *pfcsp* haplotypes of each pairing.

selection on these loci exerted by immune pressure. This large number of unique haplotypes allowed us to both identify matches between participant–mosquito pairs as well as weight the relevance of those matches for potential transmission events based on the quantity and rarity of shared haplotypes. Importantly, amplicon deep sequencing enabled this approach with its technical ability to capture minority variants within mixed infections[40] and scalability in a large field study[41]. More precise estimations of individual transmission events as well as mapping of transmission chains may require novel approaches using higher-dimensional genotyping combined with analytic models that resolve polygenomic infections. Our results highlight how integrated genetic and computational approaches can be implemented in large field studies to leverage parasite genetic diversity for investigating fundamental features of parasite epidemiology.

Using this approach, we observed that the median number of *pfcsp* haplotypes (or MOI) was much higher in mosquito infections (6) than in asymptomatic (3) or symptomatic (1) human infections (Fig. 2). This high median MOI in mosquito abdomens is surprising given that wild-caught[42] and membrane-fed[43] Anopheline mosquitoes typically have <5 oocysts, suggesting that

the high amount of genetic diversity that we observed was likely harbored by a very small number of oocysts in the collected mosquitoes. This could have resulted from the transmission to mosquitoes of cryptic haplotypes that were undetectable in asexual human infections, as has been reported with both *P. falciparum* and *P. vivax*[44], although both sample types were processed analogously and were subjected to identical haplotype quality-filtering criteria. On a related note, partial immune recognition of expressed circumsporozoite protein or apical membrane antigen-1 variants, which are expressed in the liver or blood stage, respectively, may have served to differentially limit the densities of certain variants below the limits of detection in human infections while allowing passage to and propagation in mosquitoes. Finally, given evidence that *Anopheles gambiae* can take multiple bloodmeals per gonotropic cycle[45–47] and that this behavior may be enhanced by an existing sporozoite infection of the mosquito[48], these oocysts may represent an accumulation of parasites acquired over multiple feedings on multiple days from multiple infected humans, which collectively would enhance the diversification of midgut parasites.

The finding that asymptomatic *P. falciparum* infections are the primary source of infections in mosquito vectors provides an

**Table 1 Comparison of pairings of participant and mosquito infections by symptomatic status.**

|  | Asymptomatic infections (*N* = 3012) | Symptomatic infections (*N* = 715) | *p* value |
|---|---|---|---|
| *Participant-level covariates* |  |  |  |
| Parasite density (parasites/μL), median (IQR) | 11.08 (0.96–251) | 3229.46 (505–6581) | <0.001[a] |
| Age, N (%) |  |  | <0.001[b] |
| <5 years | 326 (3.72) | 112 (15.66) |  |
| 5–15 years | 1372 (45.55) | 111 (15.52) |  |
| >15 years | 1314 (43.63) | 492 (68.81) |  |
| Mosquito abundance, N (%) |  |  | <0.001[b] |
| Low | 1564 (51.93) | 227 (31.75) |  |
| High | 1448 (48.07) | 488 (68.25) |  |
| Number of *pfcsp* haplotypes, median (IQR) | 3.00 (1.00–8.00) | 2.00 (1.00–3.00) | <0.001[a] |
| Village, N (%) |  |  | <0.001[b] |
| Maruti | 2267 (75.27) | 411 (57.48) |  |
| Kinesamo | 616 (20.45) | 208 (29.09) |  |
| Sitabicha | 129 (4.28) | 96 (13.43) |  |
| *Participant–mosquito pair-level covariates* |  |  |  |
| Probability of transmission, median (IQR) |  |  |  |
| Across all variables | 0.05 (0.00–0.16) | 0.00 (0.00–0.11) | <0.001[a] |
| Time interval[c] | 1.00 (1.00–1.00) | 1.00 (1.00–1.00) | NE |
| Distance interval | 0.67 (0.54–0.84) | 0.70 (0.53–0.80) | 1.000[a] |
| *pfcsp* haplotype sharing and prevalence[d] | 0.09 (0.00–0.24) | 0.00 (0.00–0.15) | <0.001[a] |
| For those that shared *pfcsp* haplotypes | 0.20 (0.09–0.32) | 0.19 (0.10–0.37) | 1.000[a] |
| Number of *pfcsp* haplotypes shared, median (IQR)[e] | 1.00 (0.00–2.00) | 0.00 (0.00–1.00) | <0.001[a] |
| For those that shared *pfcsp* haplotypes | 2.00 (1.00–3.00) | 1.00 (1.00–1.00) | <0.001[a] |

*IQR* interquartile range, *NE* not evaluated, *pfcsp Plasmodium falciparum* circumsporozoite protein.
[a]Wilcoxon Rank-Sum test with continuity correction and Bonferroni correction for repeated measures.
[b]Pearson's χ2 test with Bonferroni correction for repeated measures.
[c]The probability of transmission based on the time interval was set as 1 for all participant–mosquito pairings where the mosquito was collected within 7 days prior to or 14 days after the participant's infection. All pairings outside of that time interval had a probability of transmission of 0.
[d]The probability of transmission based on the *pfcsp* haplotype sharing and prevalence is shown for all pairings regardless of whether they shared the haplotypes or not.
[e]The number of *pfcsp* haplotypes shared is shown for all pairs regardless of whether they shared the haplotypes or not.

explicit rationale to target these infections in order to reduce transmission in highly endemic settings. Across sub-Saharan Africa, asymptomatic *P. falciparum* infections are highly prevalent[8,49–51]: one meta-analysis estimated a continent-wide prevalence in 2015 of 24% among just children aged 2–10 years[52]. Asymptomatic infections have been targeted in prior studies either by testing defined geographic or demographic groups (i.e., active case detection) or by foregoing testing and implementing mass-drug administration (MDA) of antimalarials[53]. Both active case detection and MDA have proven effective or been implemented in low-transmission, pre-elimination settings, where they have been recommended as interventions to accelerate progress to elimination[54]. In contrast, high-transmission settings like ours rely on bed net use, access to care, use of rapid diagnostics, and treatment with artemisinin-combination therapies for control[1]. Despite the adoption of all of these interventions in our study site, asymptomatic infections remained the major source of mosquito infections, suggesting the need for enhanced interventions. The efficacy of such interventions in high-transmission settings on the asymptomatic reservoir specifically—and on transmission reduction more generally—may be feasibly testable with novel tools to estimate transmission using serologic[55] or parasite genetic[56] measures.

Our study had several limitations. Symptomatic infections were quickly diagnosed and treated with effective therapy under our protocol, which likely reduced the duration of these infections and therefore limited their transmission potential. This access to diagnosis and treatment is higher than is generally available across sub-Saharan Africa[1], though recent reports indicate gradual improvement in quality clinical management[57]. Conversely, we may have under-detected asymptomatic infections and therefore over-represented symptomatic infections, owing either to the sparse monthly sampling for asymptomatic infections or the inability to capture transmission from symptomatic infections during their asymptomatic or pre-symptomatic phase. We expect that this would serve mainly to bias our analyses toward the null by providing relatively more opportunities for symptomatic infections to match to mosquitoes. Similarly, mosquito sampling was necessarily sparser than human sampling, precluding absolute measurement of all transmission events but allowing for relative estimations to onward transmission. We had no direct measurement of gametocytes due to the types of sample collection, precluding a direct analysis of their participation in transmission; however, we adjusted models for asexual parasite density, which has been suggested as a proxy for gametocyte density[10]. We only measured transmission directly within households and cannot capture events occurring in other settings; this limitation is mitigated by the known nocturnal feeding preference of local vectors. Finally, many infections in participants and mosquitoes had low parasite densities, which increases the risk of haplotype false discovery[58]. To mitigate this risk, we enforced stringent haplotype censoring based on read quality and haplotype abundance consistent with prior studies[58–60].

In our longitudinal study of paired participant and mosquito *P. falciparum* infections, compared to people with symptomatic malaria infections, those with asymptomatic infections were more than twice as likely to successfully transmit *P. falciparum* to *Anopheles* mosquitoes. Future studies can investigate biological and epidemiological mechanisms by which symptomaticity influences transmission as well as estimate the feasibility and efficacy of targeting asymptomatic infections as a means to reduce transmission in highly endemic settings.

## Methods
**Ethical considerations.** All adult participants provided written informed consent prior to participation. Participants between the ages of 1 and 18 years were included if their parent or legal guardian provided written informed consent.

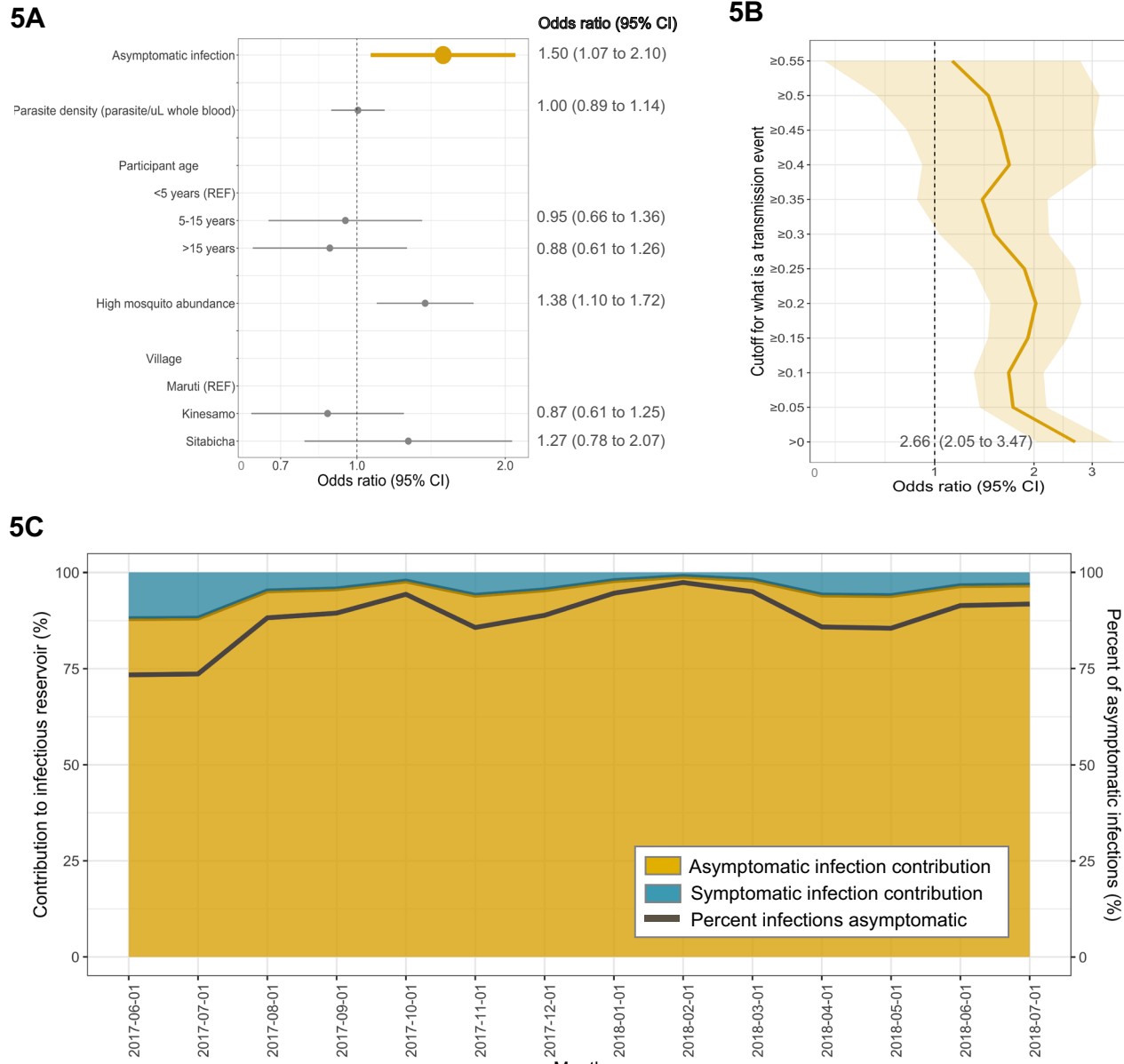

**Fig. 5 Multi-level logistic regression results for the odds of a participant-to-mosquito malaria transmission event from participants with asymptomatic compared to symptomatic infections using the *pfcsp* gene target. A** Odds ratios (ORs) of the probability of malaria transmission events from infected participants to mosquitoes ($N = 3727$ participant–mosquito pairings). ORs were computed using a multi-level logistic regression model with the probability of transmission outcome coded continuously. Values >1 indicate a factor that is associated with a greater likelihood of transmission of parasites to a mosquito, while values <1 indicate a lesser probability. Data are presented as odds ratios (dots) with 95% confidence intervals (bars). **B** ORs of the probability of transmission from infected participants to mosquitoes were re-estimated using multi-level logistic regression models with the outcome coded dichotomously ($N = 3727$ participant–mosquito pairings). Models were computed iteratively by dichotomizing the probability of transmission at increasing values from 0.00 to 0.55, thereby increasing the stringency of the definition of a transmission event. The dark yellow line indicates the OR at each dichotomized level of the probability outcome, and the shaded area indicates the 95% confidence interval around each OR. **C** The contribution to the infectious reservoir was calculated using the odds of transmission to mosquitoes from participants with asymptomatic compared to symptomatic infections each month. The yellow area indicates the percentage of asymptomatic infections that contributed to the infectious reservoir, whereas the blue area indicates the percentage of contribution of symptomatic infections. The dark gray line indicates the percentage of infections that were asymptomatic each month.

Verbal assent was also obtained from children between 8 and 18 years. The study was approved by the ethical review boards of Moi University (2017/36), Duke University (Pro00082000), and the University of North Carolina at Chapel Hill (19–1273).

**Study population and data collection.** A longitudinal cohort of households across three villages (Kinesamo, Maruti, and Sitabicha) in Bungoma county, Kenya was established in June 2017 and followed until July 2018. The three villages were

selected based on their high malaria prevalence in a previous cross-sectional study[61]. All household members in participating households over the age of 1 year were offered enrollment. Sample collection details have been reported[62]. For each participant, demographic and behavioral questionnaires were administered and dried blood spot (DBS) samples were collected every month. The DBS were tested for *P. falciparum* parasites using real-time PCR post hoc (see below), and therefore parasites detected in asymptomatic people were not treated. Participants contacted the study team at any time when experiencing symptoms consistent with malaria, at which time they were tested for malaria using a rapid diagnostic test (RDT)

(Carestart © Malaria HRP2 *Pf* from Accessbio)[63] and, if positive, treated with Artemether-Lumefantrine. DBS were also collected at the time of RDT testing. One morning each week, indoor resting mosquitoes were collected from participant households using vacuum aspiration with Prokopacks[64]. From these collections, female *Anopheles* mosquitoes were identified morphologically and transected to separate the abdomen from the head and thorax.

**Participant and mosquito sample processing.** gDNA was extracted from mosquito abdomens and DBS samples using a Chelex-100 protocol[65]. gDNA from each DBS and mosquito was tested in duplicate using a duplex TaqMan real-time PCR (quantitative PCR (qPCR)) assay targeting the *P. falciparum pfr364* motif and the human β-tubulin gene[66]. Samples were defined as *P. falciparum* positive if: (i) both replicates amplified *P. falciparum* and both Ct values were <40 or (ii) 1 replicate amplified *P. falciparum* and Ct value was <38. *P. falciparum*-positive samples were genotyped across variable segments of genes encoding the apical membrane antigen-1 (*pfama1*) and circumsporozoite protein (*pfcsp*) as previously described[22], with some additional steps taken for low parasite density samples (see Supplementary Information). Ultimately, dual-indexed libraries were prepared for both gene targets, then pooled and sequenced on an Illumina MiSeq platform (Tables S7 and S8)[67].

Sequencing reads were filtered based on read length and Phred quality scores and mapped to the 3D7 reference sequences for *pfama1* and *pfcsp*[22,68–72]. We performed haplotype inference on mapped reads using DADA2 (version 1.8) as implemented in R (version 3.6.1)[21,73]. These putative haplotypes were then further filtered in order to mitigate the risk of false discovery by removing haplotypes from a sample that met any of the following criteria: (i) supported by <250 reads within the sample, (ii) supported by <3% of the sample's total read depth, (iii) deviation from the expected nucleotide length of 300 for *pfama1* or 288 for *pfcsp*, or (iv) a minority haplotype distinguished by a one single-nucleotide polymorphism difference from another haplotype within the sample that had a read depth >8 times the read depth of the minority haplotype[58]. Finally, we removed haplotypes from the overall population if it was defined by a single variant position that was only variable within that haplotype (see Supplementary Information Figs. S1–S4).

**Exposure assessment.** The main exposure was the classification of an infection as asymptomatic or symptomatic. We defined an asymptomatic infection as a *P. falciparum* infection detected by qPCR during active case detection in a participant lacking symptoms. We defined a symptomatic infection as a *P. falciparum* infection detected by both RDT and qPCR during passive case detection in a participant with at least one malaria-like symptom. To reduce potential for exposure misclassification, individual asymptomatic and symptomatic infections were excluded from the analysis if they occurred within 14 days of taking study-prescribed antimalarials for a symptomatic infection.

**Within-participant modeling of transmissibility.** To assess the likelihood of transmission by symptomatic status, we compared the proportion of mosquitoes that shared a haplotype between a participant's asymptomatic and symptomatic infections. To do so, we included only participants who had at least one asymptomatic and one symptomatic infection. We then paired each participant's infection events with all mosquitoes that were collected within 3 km as well as between 7 days prior to and 14 days following the participant infection, in order to constrain the search space for plausible transmission events to within time and distance parameters that are consistent with parasite and mosquito biology. For each infection, we computed the proportion of participant–mosquito pairings that shared at least one haplotype and did so separately using either *pfcsp* or *pfama1* haplotypes.

We assessed the statistical significance of differences in these proportions between asymptomatic and symptomatic infections using a multi-level logistic regression model (Eq. 1):

$$\ln\left(\text{Transmission event}_{ij}\right) = \alpha_i + \alpha_j + \beta_1 \text{Asymptomatic infection}_{ij} \\ + \beta_2 \text{Parasite density}_{ij} + \beta_3 \text{Mosquito abundance}_{ij} + \epsilon_{ij} \quad (1)$$

This model included a random intercept at the participant level ($\alpha_i$) and one at the household level ($\alpha_j$) to account for repeated measures of participants clustered in households throughout the study. The model included covariates for parasite density (in parasites/μL in the participant samples) and mosquito abundance (expressed as the total number of female *Anopheles* mosquitoes collected within the week following the participant infection as <75 mosquitoes or ≥75 mosquitoes). The cutoffs for the number of mosquitoes chosen to represent mosquito abundance was determined by a functional form assessment and known malaria seasonality.

**Probabilistic modeling of transmission across all participants.** For a more comprehensive measure of transmission across all participants, we created a probabilistic model to estimate the probability that a shared haplotype between a participant and a mosquito represented a *P. falciparum* transmission event [$P(\text{TE}_{\text{all}})$]. $P(\text{TE}_{\text{all}})$ was estimated for the pairing of each infected participant with an infected mosquito on the basis of three distinct features: (i) the time interval between the participant's infection and mosquito collection [$P(\text{TE}_t)$], (ii) the

distance between the household of the participant and the household where the mosquito was collected [$P(\text{TE}_d)$], and (iii) the prevalence and number of parasite haplotypes shared [$P(\text{TE}_h)$]. For each pairing, we calculated these terms and then multiplied them to estimate $P(\text{TE}_{\text{all}})$. $P(\text{TE}_{\text{all}})$ values were computed independently using the *pfama1* or *pfcsp* haplotypes.

*Probability of participant-to-mosquito transmission over time.* $P(\text{TE}_t)$ was defined as the probability of participant-to-mosquito parasite transmission as a function of the time interval between specimen collections. The rationale for this term was that participant-to-mosquito transmission could only occur within a certain time window based on the mosquito lifespan and parasite life cycle[74,75]. $P(\text{TE}_t)$ was assigned as 1 if a mosquito was collected within a 21-day window of the participant infection, spanning 7 days before the participant infection and 14 days after (Fig. S5). This range only allowed participant-to-mosquito malaria transmission to be captured, following infections from participants to mosquito abdomens. If the mosquito was collected outside of this window, $P(\text{TE}_t)$ was set to 0. A sensitivity analysis was conducted to assess how differences in the time window chosen affected results, expanding the time window to allow mosquitoes to be collected up to 30 days after the participant's infection (Fig. S6).

*Probability of participant-to-mosquito transmission over distance.* $P(\text{TE}_d)$ was defined as the probability of participant-to-mosquito parasite transmission as a function of Euclidean distance between specimen collections. This term was included to restrict participant–mosquito pairs to only be considered as a possible transmission event within a reasonable distance for a mosquito to fly. $P(\text{TE}_d)$ was calculated using a modified negative exponential distribution previously observed in a study tracking *Anopheles* mosquito movement (Eq. 2 and Fig. S7)[76]:

$$P(\text{TE}_d) = e^{-3d} \quad (2)$$

For example, by 0.66 km from the participant, which was the maximum distance blood-fed *Anopheles* mosquitoes were observed to fly in a Kilifi study[77], the probability of transmission was already low (14%) and at 3 km it had dropped to 0% entirely. A sensitivity analysis was conducted to evaluate how changing the distance between specimen collection to allow collection at a distance >3 km influenced results (Fig. S8). We also compared the number of *pfcsp* haplotypes shared within 3 km compared to at a distance of >3 km (Fig. S9).

*Probability of participant-to-mosquito transmission over haplotypes.* $P(\text{TE}_h)$ was defined as the probability that a shared haplotype represented a participant-to-mosquito parasite transmission as a function of the number of shared haplotypes and the population prevalence of each shared haplotype. The premise of the calculation of $P(\text{TE}_h)$ was that the probability that haplotype sharing represented a transmission event increased with a higher number of haplotypes shared as well as the rarity of those haplotypes across the study population. $P(\text{TE}_h)$ was calculated independently for *pfama1* and *pfcsp* haplotypes using Eq. 3:

$$P(\text{TE}_h) = \left(1 - \prod_{n=1}^{s} \text{PopPrev}_n^{1/3}\right)\left(\frac{s}{\text{MOI}_i}\right) \quad (3)$$

where:

1. *s* indicates the number of haplotypes of a gene target (*pfcsp* or *pfama1*) that are shared between components of the pair;
2. PopPrev indicates the prevalence of the haplotype across the entire study population, calculated by dividing the number of samples with that haplotype by the total number of samples in the study. PopPrev is calculated from 1 to *s*, where *s* is the total number of shared haplotypes between the participant and mosquito pair. PopPrev for each haplotype is rescaled by taking the cubed root, as it is highly right skewed; and
3. $\text{MOI}_i$ is the participant's MOI, represented by the number of *unique gene* haplotypes observed in the participant's infection (*i*).

We applied the term $\left(\frac{s}{\text{MOI}_i}\right)$ in order to mitigate the risk of biasing $P(\text{TE}_h)$ toward larger values in participants with high MOI values. If no haplotypes were shared between the participant and mosquito pair, $P(\text{TE}_h) = 0$. $P(\text{TE}_h)$ was calculated independently for *pfcsp* and *pfama1*. A sensitivity analysis was conducted comparing $P(\text{TE}_h)$ calculated independently for each gene target to $P(\text{TE}_h)$ calculated using both gene targets (Fig. S10).

*Probability of participant-to-mosquito transmission over all variables.* Individual terms for $P(\text{TE}_t)$, $P(\text{TE}_d)$, and $P(\text{TE}_h)$ were combined into a final estimate of the probability of transmission [$P(\text{TE}_{\text{all}})$]. For participant–mosquito pairs that had probability values >0 of $P(\text{TE}_t)$, $P(\text{TE}_d)$, and $P(\text{TE}_h)$, $P(\text{TE}_{\text{all}})$ was calculated using Eq. 4:

$$P(\text{TE}_{\text{all}}) = P(\text{TE}_t) * P(\text{TE}_d) * P(\text{TE}_h) \quad (4)$$

If $P(\text{TE}_t) > 0$ and $P(\text{TE}_d) > 0$ but $P(\text{TE}_h) = 0$, $P(\text{TE}_{\text{all}}) = 0$. $P(\text{TE}_t)$, $P(\text{TE}_d)$, and $P(\text{TE}_h)$ were rescaled to the range 0–1 to be comparable when multiplying.

**Statistical analysis**. To estimate the probability of a participant-to-mosquito transmission event using the probabilistic method for participant-to-mosquito transmission [$P(\text{TE}_{all})$] across time, distance, and the haplotypes shared, we compared values between participants with asymptomatic and symptomatic infections using a multi-level logistic regression model (Eq. 5).

$$
\begin{aligned}
\ln\left(\text{Transmission event}_{ij}\right) = {} & \alpha_i + \alpha_j + \beta_1 \text{Asymptomatic infection}_{ij} \\
& + \beta_2 \text{Parasite density}_{ij} + \beta_3 \text{Age5to15}_{ij} + \beta_4 \text{Age15over}_{ij} \\
& + \beta_5 \text{Mosquito abundance}_{ij} \\
& + \beta_6 \text{Kinesamo village} + \beta_7 \text{Sitabicha village} + \epsilon_{ij}
\end{aligned}
\tag{5}
$$

We included a random intercept at the participant level ($i$) to account for repeated measures for participants who experienced multiple malaria infections (asymptomatic or symptomatic) throughout the study. To consider different transmission intensities between households, we included a random intercept at the household level ($j$). We controlled for confounding covariates that we identified in a directed acyclic graph (Fig. S11) and performed functional form assessments on continuous variables prior to inclusion (See Supplementary Information, Tables S3–S5). The final model included covariates for village, parasite density in the participant samples in parasites/µL (linear), participant age at study enrollment (categorized: <5 years, 5–15 years, >15 years), and mosquito abundance (expressed as the total number of female *Anopheles* mosquitoes collected within the week following the participant infection as <75 mosquitoes or ≥75 mosquitoes). To reduce skew for the multi-level model, parasite density was centered and rescaled to have a mean of 0.

Prior to modeling, differences in model covariates across symptomatic status were assessed using Wilcoxon Rank-Sum test with continuity correction for continuous variables and the Pearson's $\chi^2$ test for categorical variables. Differences in MOI across sample types were calculated using pairwise Wilcoxon Rank-Sum test with the Bonferroni correction for multiple comparisons. All $p$ values obtained from the bivariate tests were adjusted using the Bonferroni correction to account for repeated measures across participants of up to 14 infections, which was the maximum number of infections observed in any participant during study follow-up.

We conducted a sensitivity analysis of different coding choices for the probabilistic combination of time, distance, and haplotypes [$P(\text{TE}_{all})$]. We re-computed the logistic regression model specified above testing variable cutoffs for the binary coding of the outcome variables. Cutoffs for what was considered a participant-to-mosquito malaria transmission ranged from 0.00 to 0.55 due to sparse data restrictions >0.55. The model contained the same covariates and random effects as the model in Eq. 5.

**Contribution to infectious reservoir of asymptomatic infections**. The contribution to the infectious reservoir made by asymptomatic infections was calculated using the OR estimate obtained from the probabilistic method for participant-to-mosquito transmission [$P(\text{TE}_{all})$] across time, distance, and the haplotypes shared. The OR for the binary coding of $P(\text{TE}_{all})$ was used, where any value of $P(\text{TE}_{all})$ indicated a participant-to-mosquito transmission event. The contribution to the infectious reservoir ($C_{at}$) was estimated using Eq. 6[17,78]:

$$
C_{at} = \frac{P_{at}I_a}{P_{at}I_a + P_{st}I_s} * (100)
\tag{6}
$$

where:

1. $P_{at}$ represents the proportion of all infections that were asymptomatic ($a$) during each month ($t$) of follow-up;
2. $P_{st}$ represents the proportion of all infections that were symptomatic ($s$) during each month ($t$) of follow-up;
3. $I_a$ indicates the likelihood a mosquito was infected by someone with an asymptomatic ($a$) infection. $I_a$ was calculated using the OR obtained from the multi-level logistic regression model that estimated the probability of participant–mosquito transmission with $P(\text{TE}_{all})$ coded binarily: $\text{OR} = \frac{I_a}{I_s}$. Model random effects and covariates were previously described in Eq. 5. $I_a$ did not vary across months; and
4. $I_s$ represents the likelihood a mosquito was infected by someone with a symptomatic ($s$) infection. $I_s$ was calculated as follows: $I_s = 1 - I_a$. $I_s$ did not vary across months.

A cumulative value of $C_{at}$ was calculated across the entire follow-up period and represented by $C_a$. The upper and lower limits of the 95% CI for $C_a$ were calculated using the upper and lower limits for the 95% CI from the estimated OR. All statistical analyses were performed using R (version 3.6.1)[73].

**Reporting summary**. Further information on research design is available in the Nature Research Reporting Summary linked to this article.

## Data availability

The datasets generated during the current study are available in the GitHub repository: https://github.com/duke-malaria-collaboratory/human_mosquito_manuscript.git. The sequence data analyzed during the current study are available from NCBI (PRJNA646940).

## Code availability

R code used for analyses is available on GitHub: https://github.com/duke-malaria-collaboratory/human_mosquito_manuscript.git.

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

## Acknowledgements

We thank the field technicians in Webuye for their engagement with the study participants: I. Khaoya, L. Marango, E. Mukeli, E. Nalianya, J. Namae, L. Nukewa, E. Wamalwa, and A. Wekesa. In addition, we appreciate the study development and coordinating skills of J. Mangeni and J. Kipkoech Kirui, the analytic advice offered by J. Edwards and M. Emch (both from the University of North Carolina at Chapel Hill), and the assistance with laboratory sample and data processing by V. Liao, A. Nantume, S. Kim, and J. Saelens (each of Duke University). This work was supported by NIAID (R21AI126024 to W.P.-O. and R01AI146849 to W.P.-O. and S.M.T.). A.P.W. was supported by a Career Award at the Scientific Interface by the Burroughs Wellcome Fund. Ultimately, we are indebted to the study households for their participation in this study.

## Author contributions

K.M.S., S.M.T., and W.P.-O. conceptualized and designed the study with additional input from A.O., S.R.M., B.W.P., and A.W. L.A. and A.O. implemented field collections of human and mosquito specimens. E.F. performed molecular sample processing. K.M.S. carried out the data analysis with additional input from S.M.T., W.P.-O., A.W., S.R.M., L.A., A.O., and B.W.P. K.M.S. wrote the first draft of the manuscript. All authors contributed to the manuscript and approved the final draft.

## Competing interests

The authors declare no competing interests.
