## [Peer Review File · Nature Communications]

Reviewers' Comments:

Reviewer #1:

Remarks to the Author:

The submitted manuscript, "Genotyping of cognate *Plasmodium falciparum* parasites in naturally-infected human and mosquito hosts to estimate the contribution of asymptomatic infections to onward malaria transmission" describes dense sampling of symptomatic and asymptomatic malaria infections, coupled with mosquitos from the same household. The authors perform deep haplotype sequencing of two highly polymorphic genes, and build a statistical model to infer the relative contribution of asymptomatic and symptomatic cases to the overall transmission rate. Understanding the reservoir of ongoing malaria transmission is a major question in malaria research and is critical to deciding how we can target valuable resources to better control malaria. The authors are bringing great tools to improve our understanding of this asymptomatic reservoir. This is a unique dataset, which was undoubtedly a major effort to collect.

I have some fundamental questions about the paper which are, I feel, critical to understanding whether the results robustly estimate the relative contribution of each infection class to transmission.

What are the expectations of actually capturing transmission events at all? I struggled to comprehend this. Is much known about where people transmit (and acquire) their infections in this area? Mosquitos within homes may be an excellent surrogate for some transmission events, especially for individuals who spend times of day with elevated transmission rates in their homes. For others it may be futile, for instance, if school age children transmit infections at school the chances of capturing the precise mosquitos which fed on them is small and could underpower the symptomatic group. In brief, to what extent would you reasonably expect the mosquito sampling performed to be representative about where transmission actually occurs?

Lines 99-114: The result here is that there is a greater chance of sharing a parasite haplotype with an infected mosquito for asymptomatic individuals compared to symptomatic individuals. In lines 97-98 the MOI for asymptomatic individuals is 3, vs 1 in symptomatic individuals. Given that there are more haplotypes in one class this would seem to confound the analysis. This same issue would appear to apply to the analysis described in lines 123-124, 150-157.

I was very intrigued by the high MOI in mosquitos. To my mind, these either derived from haplotypes which were undetectable in an individuals blood (as has been seen in controlled feeding experiments) or they result from interrupted feeding on multiple individuals. To what extent does this data reveal why these mosquitoes have a high MOI?

To my reading the transmission model was derived specifically for the data presented here. The inclusion of modelling in the paper is a major strength. Some guidance in interpretation would have aided me. For instance, does $P(TE_{all})$ of 0.05 equate to a 5% chance this was an actual transmission event. If so, does this mean there were essentially no transmission events captured? Additionally, I did not see details on how the model was tested for accuracy (for instance on simulated data), was this performed?

The loci genotyped in this study are under strong pressure from the immune system. This may impact the estimates of MOI using these markers, and may contribute to differences between symptomatic and asymptomatic individuals as they acquire allele specific immunity. Further discussion of why these markers were used would be beneficial. Additionally, in the discussion you state that *pfama1* and *pfmsp* are unlinked. At a population level this is true. Though, for a true transmission event they are in perfect linkage (as is the whole genome). I would suggest that identifying mosquito-human pairs which share haplotypes at both loci are highly informative about genuine transmission events. If feasible, genotyping additional neutral loci in mosquito-human pairs with varying $P(TE_{all})$ estimates could be more conclusive.

Ian Cheeseman

Reviewer #2:

Remarks to the Author:

The authors have carried out an impressive longitudinal study of the genetics of malaria parasites in a human population in Kenya, and in mosquitoes sampled by resting collections from the same locations. The possibility of comparing the genetics of the parasites from a large number of mosquitoes with those in the humans who transmitted these, makes this an unusual and potentially valuable field study that could address previously unanswered questions in malaria genetics. For instance, the extent to which the bloodmeal represents a bottleneck in propagating parasite genetic diversity is unknown. How does MOI in the human host translate into heterozygosity in oocysts and genetic variation in sporozoites?

However, instead of directly exploiting their genetic data, the authors present a comparison of estimated infectiousness of symptomatic and asymptomatic infections, claiming that this fills a research gap. However, there have been many studies of infectiousness in malaria in nature, mostly using membrane feeding assays which provide a rather direct means of assessing the extent to which different individuals transmit. In contrast to the authors' claim (Lines 25-26) the contribution of asymptomatic infections to transmission has been recognised for many decades. The authors have access to this literature, much of which, except the most recent studies, is reviewed in reference [6], so it is not obvious why they have cited only a heterogeneous collection of other studies mostly focussing on other topics (references 14-18) as examples of studies of infectiousness of asymptomatic infections.

The hypothesised relationship between the authors' 'transmission events' and actual transmission events is very indirect. Nevertheless, on lines 26-27 the authors claim their very indirect estimates are 'direct'. This seems to be a misnomer. Any comparison of parasites found in wild mosquitoes with those found in human hosts must address the challenge that it is impossible to capture most of the mosquitoes that are biting humans, so the mosquitoes represent a small, (and possibly unrepresentative) sample, so at best the study can investigate statistical correlations. This leads to challenges with potential confounding factors:

- Untreated malaria infections are typically symptomatic for a period early in the infection, followed by a long asymptomatic tail. Because of the lag time in the human host between densities of blood stage infections and infectiousness, (corresponding to the duration of gametocytogenesis) it is to be expected that untreated infections are most infectious in the period after a symptomatic attack. It follows that symptomatic and asymptomatic infections, detected at surveys, can be genetically identical with the difference just reflecting the time point at which the sample was drawn. So the authors 'symptomatic' infections were undoubtedly 'asymptomatic' at another (unsampled) time point.
- The temporal effects that are reported (lines 158-164) could be secondary to any of a number of seasonal epidemiological trends, of which the seasonal pattern of pathology is only one.
- Before using of genetic differences as a marker of whether the source infection is symptomatic, the investigators need to establish that there are genetic differences between these categories in the markers in the study. However, other studies of Pfcpr and Pfama-1 do not suggest that genotypes at these loci encode virulence factors.
- Any relationship of parasite genetics with disease could easily be confounded, for instance by an effect of age. The age of the host is known to affect infectiousness (most likely via parasite densities). It also affects the ratio of asymptomatic/symptomatic periods of infection. The age of the host is also likely to differentially affect the densities (as the authors recognise, lines 126-128) and hence infectiousness of different parasite genotypes independently of any effect of symptomatic status.

A further set of problems arise because of the structure of the model linking the mosquito and human data, which very likely introduces biases:

- The different quantities $P(\text{TEt})$, $P(\text{TEd})$ and $P(\text{TEh})$ (see methods and Table 1) do not behave like

probabilities (as is illustrated by the need to consider truncation and re-scaling in order to constrain $P(TE_{all})$ to be within the allowable range (equation 4). It is not clear what are the implications of this for the estimates of relative infectiousness of asymptomatic infections, but a model that considered relative rates, rather than probabilities that are not probabilities, would be more convincing. - The 'pruning' of incompatible genetic types from the analysis (lines 118-120) also seems likely to introduce bias, though it is not clear in what direction. The fact that a genetic type was not detected does not mean that it was not present.

- The assumption of a log-linear relationship (equation 1) needs to be justified. Both previous empirical studies and theoretical models suggest that the relationship of infectiousness to parasite densities is non-linear. This may be critical for the analysis because parasite densities are a major potential confounder.

- The inclusion of both passively and actively detected cases among symptomatic infections may be a source of bias since the time at risk while symptomatic is overrepresented in the dataset.

Further issues with the paper include:

It is not clear whether the parameter MOI_i refers to the unique haplotypes at a single time point or among the complete set of samples from the individual.

Line 43-44. Reference [1] indicates a gradual decrease in malaria incidence since 2010, not a plateau. Line 95. The finding that pfcsr multiplicity of infection was higher for mosquitoes than for human hosts is very surprising. In most studies, the majority of infected mosquitoes contain only single oocysts, which should correspond to MOI of two or less (and mosquitoes are unlikely to be superinfected). Do the authors have any explanation for this?

Reviewer #3:

Remarks to the Author:

Review: Genotyping of cognate *Plasmodium falciparum* parasites in naturally-infected human and mosquito hosts to estimate the contribution of asymptomatic infections to onward malaria transmission

This is a well written paper that attempts to answer important questions in malaria epidemiology on the relative contribution of asymptomatic and symptomatic individuals to onwards malaria transmission. The dataset is very novel as it is based on natural infections in the field rather than membrane feeding in a laboratory setting as has been used before, meaning new methods were needed to link infected mosquitoes to infected humans. I feel the authors have developed appropriate methods that well describe a complicated dataset to reach interesting and robust conclusions. My main comments are related to things that could be clarified and a few validations/sensitivity analyses.

Major comments

Given the longitudinal nature of the dataset, is it possible to estimate the mean duration of an asymptomatic infection? Or even the mean duration of a given haplotype infection within an individual? Given the very quick treatment of symptomatics in this study, it would mean you could look at how the relative contribution might vary in more realistic treatment scenarios. Also given that in reality a lot of symptomatic individuals never seek treatment and just transition to becoming asymptomatic, how does that fit in with this paper? That would suggest advocating for better treatment access rather than MDA approaches I think. I don't want to be the reviewer that suggests you write a whole different paper! But maybe useful to think about what value these bits of information would add

The approach for calculating $P(TE_{all})$ makes sense, but I was wondering if you could do a loose validation by looking at counterfactuals, i.e. is there an increased number of shared haplotypes in human-mosquito samples that are <3km compared to >3km, is there an increased number of shared haplotypes in samples within your infection window compared to outside it. I think seeing these data would reassure me that you are indeed picking up truly related infections (I'd recommend this as an addition to the supplement, not the main text).

L34-38 : this sentence isn't very clear – I think the start 'using these haplotypes...' is too unspecific, meaning it's not clear exactly what you're comparing. I'd recommend very briefly describing the method – i.e. 'Using haplotypes to probabilistically link infections in humans and mosquitoes...' or something. And then, 'among individuals that had both symptomatic and asymptomatic infections during the course of the study, we found that asymptomatic infections had double the odds of infecting a mosquito compared to symptomatic infections'.

L97 : Interesting that MOI was higher in asymptomatic infections – is this because 1 strain is swamping the reads in symptomatic infections? Are there instances where individuals have asymptomatic infection in the sample prior to their symptomatic episode? Are the haplotypes of the asymptomatic infection the same as the symptomatic haplotype. If an individual had multiple haplotypes in the same before getting sick, it'd be unlikely they'd cleared them all in <1 month. L101 : are these 65 individuals a biased sample and therefore unrepresentative? i.e. how does the mean age compare to the whole population, are they a group of individuals that just get bitten more? How does # infections in this group compare to the population as a whole?

L108 : in the logistic regression, did you account for the fact that there were multiple timepoints from the same individual? Did you account for the frequency of the haplotype in the population? Did you transform the parasite density data (i.e. log10) as otherwise models can get really skewed by very high numbers.

L159-160: normally 'prevalence of asymptomatic infections' refers to proportion of whole population that have an asymptomatic infection. I think you mean something like 'monthly proportion of all infections that are asymptomatic'.

L242 : I disagree that this access is generalizable – from the last world malaria report "Based on 19 household surveys conducted in sub-Saharan Africa between 2015 and 2017, the percentage of children aged under 5 years with a fever who received any antimalarial drug was 29% (IQR: 15–48%)." – and given that symptomatic kids are more likely to get treatment than symptomatic adults, I think the population-level figure will be even lower than this 29% - a long way away from the likely ~90% in this study. Also here, referring back to my first comment in major revisions – about how in reality some symptomatic individuals never seek treatment and just become asymptomatic over time.

L 357 – why 14 days after – if the individual became asexual parasite positive on the day of screening, they might not even develop gametocytes for 10 days or so, so I think 14 days might be too stringent – did you look at any sensitivity around this value?

L402 – by rescaling the probabilities, aren't you in effect just assign arbitrary weights to each factor? You mention you do a sensitivity analyses but I couldn't find the details? I'd recommend logging the equation and then assigning weights to each component and varying the beta values here to see if that varies the results much

i.e. $\log(P(\text{TE}_{\text{all}})) = \beta_1 * \log(P(\text{TE}_{\text{t}})) + \beta_2 * \log(P(\text{TE}_{\text{d}})) + \beta_3 * \log(P(\text{TE}_{\text{h}}))$

Minor comments

L81/Fig 1 : I was initially confused about the fact there were 239 participants but the bars in fig 1, row 3 only had bars 60 high, but figured it out – maybe useful to add a note on the scheduling of ACD visits in the villages to clarify this?

L89 : can you be explicit about why the majority of results for pfama1 are in the supplement? Is this known to be a less good gene for this type of analyses? Or were results just consistently less significant?

L130 : It would be interesting to see the full range of values as well as the IQR here

L179 : 'consonant' should be 'consistent'

L205 – not sure either of the papers referenced here really look at 'predictive' models. There is another Okell paper looking at modelling MSAT/MDA approaches – do you mean this one

<https://journals.plos.org/plosone/article?id=10.1371/journal.pone.0020179>

L274 – I think useful to highlight that PCR positive asymptomatic individuals were not treated

L348 – clearer to just say 'multiplied' rather than 'aggregated'

L370 – clearer to just say the number? 'For example, by 0.66km from the human host, the probability is already low (14%).

L409 – define i and j

L411 – do the village dummy variables need i and j indexes?

L441 – define a and t in the definition of the contribution to the infectious reservoir

L445 – is this proportion of all infections that were symptomatic? A bit unclear

Fig 1 – add in prevalence lines on a second y-axis for all three graphs?

Fig 2 – hard to see the small bars – add $n=XX$ for each bar in left and right margins of graph?

Fig 4D – would this be easier to interpret as a histogram?

Reviewer # 1

1. What are the expectations of actually capturing transmission events at all? I struggled to comprehend this. Is much known about where people transmit (and acquire) their infections in this area? Mosquitos within homes may be an excellent surrogate for some transmission events, especially for individuals who spend times of day with elevated transmission rates in their homes. For others it may be futile, for instance, if school age children transmit infections at school the chances of capturing the precise mosquitos

which fed on them is small and could underpower the symptomatic group. In brief, to what extent would you reasonably expect the mosquito sampling performed to be representative about where transmission actually occurs?

RESPONSE: Agreed. We failed to highlight that, in contrast to other vector-borne diseases, the Anopheline vectors overall, and specifically the species in W Kenya, are mostly obligate endophilic and endophagic feeders, and this fact enables a household study like ours – with early-morning collection of indoor-resting mosquitos – to capture a plausibly sizable portion of overall transmission. We have directly addressed this now in the Introduction: “In these households, we collected cognate infections in both people and indoor-resting Anopheline mosquitos, under the premise that, owing to the endophilic and endophagic preferences for feeding by the principal vectors *A gambiae* and *A funestus*, household transmission would be both measurable and substantial.” We have also added to the Discussion: “We only measured transmission directly within households, and cannot capture events occurring in other settings; this limitation is mitigated by the known nocturnal feeding preference of local vectors.”

2. Lines 99-114: The result here is that there is a greater chance of sharing a parasite haplotype with an infected mosquito for asymptomatic individuals compared to symptomatic individuals. In lines 97-98 the MOI for asymptomatic individuals is 3, vs 1 in symptomatic individuals. Given that there are more haplotypes in one class this would seem to confound the analysis. This same issue would appear to apply to the analysis described in lines 123-124, 150-157.

RESPONSE: We struggled with this very issue and ultimately incorporated MOI into the estimates of P(TE), specifically into the term $P(TE_h)$ as described in Equation 3, as a correction factor to prevent a higher likelihood of matching simply by virtue of harboring more haplotypes and therefore skewing the P(TE) values towards those with higher MOIs. The specific term in the equation is denoted (s / MOI_i) , where s is the number of haplotypes shared between the pair, and MOI_i is the participant's multiplicity of infection. Prior to the inclusion of this particular term, we observed that P(TE) values increased as a function of the MOI of the human infection:

This observed relationship suggested that P(TE) values were too highly dependent on MOI, of which larger values would simply allow more chances of matching. Following inclusion of the above term, P(TE) values no longer increased as a function of MOI:

Here, the MOI values have not changed but their relationship with the y-values has, and these no longer substantially increase as a function of MOI. Effectively, the term penalizes P(TE) values for the non-sharing of haplotypes between humans and mosquitos, to offset the potential increases by chance in high MOI infections.

3. I was very intrigued by the high MOI in mosquitos. To my mind, these either derived from haplotypes which were undetectable in an individuals blood (as has been seen in

controlled feeding experiments) or they result from interrupted feeding on multiple individuals. To what extent does this data reveal why these mosquitoes have a high MOI?

RESPONSE: Owing to this comment and to #22 below, we have added this paragraph to the Discussion: “Using this approach, we observed that the median number of *pfcs*p haplotypes (or MOI) was much higher in mosquito infections (6) than in asymptomatic (3) or symptomatic (1) human infections (Figure 2). This high median MOI in mosquito abdomens is surprising given that wild-caught [Rosenberg R Trends Parasitol 2008] and membrane-fed [Gneme A et al. Malaria J 2013] Anopheline mosquitos typically have < 5 oocysts, suggesting that the high amount of genetic diversity that we observed was likely harbored by a very small number of oocysts in the collected mosquitos. This could have resulted from the transmission to mosquitos of cryptic haplotypes that were undetectable in asexual human infections, although both sample types were processed analogously and were subjected to identical haplotype quality filtering criteria. Additionally, the diploid nature of oocyst forms would be expected to increase diversity compared with the haploid asexual forms, although without a known recombination hotspot within our sequenced targets it is not likely that recombination on the timescale of our study enhanced the observed parasite diversity in mosquitos. Finally, given that *Anopheles gambiae* commonly takes multiple bloodmeals per gonotrophic cycle,[Scott TW and Takken W Trend Parasitol 2012] these oocysts may represent an accumulation of parasites acquired over multiple feedings on multiple days from multiply-infected humans, which collectively would enhance the diversification of midgut parasites.”

4. To my reading the transmission model was derived specifically for the data presented here. The inclusion of modelling in the paper is a major strength. Some guidance in interpretation would have aided me. For instance, does $P(TE_{all})$ of 0.05 equate to a 5% chance this was an actual transmission event. If so, does this mean there were essentially no transmission events captured? Additionally, I did not see details on how the model was tested for accuracy (for instance on simulated data), was this performed?

RESPONSE: We have added to the Results: “ $P(TE_{aji})$ represented a relative likelihood that a human and mosquito pair that shared parasite haplotypes represented a transmission event observed in the study and should be interpreted relative to other $P(TE_{aji})$ values.” To test the model for accuracy and assess the dependency of our findings on our model assumptions, we’ve added to the Supplement the results of several sensitivity analyses. Specifically, we’ve included a sensitivity analysis in which the search space for matching mosquitos varied over the expected time interval (**S6 Fig**):

“S6 Fig. Sensitivity analysis for probability of transmission over time. A sensitivity analysis was performed to understand the influence of the time window allotted for mosquito matching to human infections (y-axis) on the outcome of the relative likelihood of transmission from an asymptomatic infection. To do so, the multi-level logistic regression model comparing probabilities of transmission events $P(TE)$ between asymptomatic and symptomatic infections was computed using time windows for mosquito collection in which the window consistently began 7 days prior to the human infection (+7) but the interval after the human infection was allowed to vary between 30 and 14 days (i.e. -30 to -14 days).. The pfmsp haplotypes were used for this sensitivity analysis.”

Next, in which the search space for mosquitoes was then allowed to vary for the expected distance interval (S8 Fig):

“S8 Fig. Sensitivity analysis for probability of transmission over distance. A sensitivity analysis was done to comparing different distance cutoffs for the probability of transmission and the effect on the relationship observed. The multi-level logistic regression model was reran comparing the probability of transmission to mosquitoes across participants with asymptomatic compared to symptomatic infections using each distance cutoff. Each maximum distance cutoff is shown on the y-axis and the associated odds ratio for transmission to mosquitoes on the x-axis. The pfcsp haplotypes were used for this sensitivity analysis.”

And finally in an ensemble model that analyzes *pfmsp* and *pfama1* concurrently (**S10 Fig**):

“S10 Fig. Sensitivity analysis for probability of transmission over haplotypes. A sensitivity analysis was done using a different coding for the P([TE] h) term where it was no longer calculated separately for pfama1 and pfcsp but instead calculated as a combined value using both pfama1 and pfcsp haplotypes. The multi-level logistic regression model was recomputed comparing the probability of transmission to mosquitoes across participants with asymptomatic compared to symptomatic infections.”

5. *The loci genotyped in this study are under strong pressure from the immune system. This may impact the estimates of MOI using these markers, and may contribute to differences between symptomatic and asymptomatic individuals as they acquire allele specific immunity. Further discussion of why these markers were used would be beneficial.*

RESPONSE: We have added to the Results: “*Pfcsp* and *pfama1* were selected owing not to phenotypes associated with their protein products but rather to their sequence diversity, which enables capture of diverse parasite strains and matching

strains between hosts.” We have also added to the Discussion: “...high diversity of *pfama1* and *pfmsp* haplotypes in our study site, likely the result of strong balancing selection on these loci exerted by immune pressure.”

6. *Additionally, in the discussion you state that pfama1 and pfmsp are unlinked. At a population level this is true. Though, for a true transmission event they are in perfect linkage (as is the whole genome). I would suggest that identifying mosquito-human pairs which share haplotypes at both loci are highly informative about genuine transmission events.*

RESPONSE: We conducted the primary analyses separately for the two gene targets because we observed differences in the number of haplotypes observed across *pfama1* and *pfmsp* targets that could confound results. As a sensitivity analysis to see how combining the haplotypes across *pfama1* and *pfmsp* affected model results, we re-computed the probabilistic model using a new value for $P(TE_h)$ made from both *pfama1* and *pfmsp*. Results have been added to the supplementary material (Fig S10) and some text describing this sensitivity analysis added to the Methods section (L479-481): “ $P(TE_h)$ was calculated independently for *pfmsp* and *pfama1*. A sensitivity analysis was conducted comparing $P(TE_h)$ calculated independently for each gene target to $P(TE_h)$ calculated using both gene targets (S10 Fig).”

7. *If feasible, genotyping additional neutral loci in mosquito-human pairs with varying $P(TE_{all})$ estimates could be more conclusive.*

RESPONSE: Unfortunately, we simply have no remaining genomic DNA from the infected mosquitos, because of the very limited number of parasites in these samples and the inability to preserve any portion of the sample during processing.

Reviewer #2

0. *... the extent to which the bloodmeal represents a bottleneck in propagating parasite diversity is unknown. How does MOI in the human host translate into heterozygosity in oocysts and genetic variation in sporozoites?*

RESPONSE: We're also interested to explore parasite diversity and population bottlenecks with these data, and are planning to do so in a separate report. The focus here was specifically on using the genotypes as markers of parasite strains for the purpose of addressing a more applied issue of parasite transmission.

8. *However, instead of directly exploiting their genetic data, the authors present a comparison of estimated infectiousness of symptomatic and asymptomatic infections,*

claiming that this fills a research gap. However, there have been many studies of infectiousness in malaria in nature, mostly using membrane feeding assays which provide a rather direct means of assessing the extent to which different individuals transmit. In contrast to the authors' claim (Lines 25-26) the contribution of asymptomatic infections to transmission has been recognised for many decades. The authors have access to this literature, much of which, except the most recent studies, is reviewed in reference [6], so it is not obvious why they have cited only a heterogeneous collection of other studies mostly focussing on other topics (references 14-18) as examples of studies of infectiousness of asymptomatic infections.

RESPONSE: We did not adequately acknowledge both the findings of prior studies as well as the limitations to their interpretation in the Introduction. We have added to this section: "Several studies have compared the transmission potential of asymptomatic and symptomatic *P. falciparum* infections to mosquitoes and have generally confirmed that such infections are transmissible [14–18]. However, the small sample sizes and the use of experimental approaches using artificial membrane feeding by laboratory-reared mosquitos limit generalizability by failing to capture variations in human activity, vector complexity and behavior, and parasite biology that influence transmissibility in natural settings."

10. *The hypothesised relationship between the authors' 'transmission events' and actual transmission events is very indirect. Nevertheless, on lines 26-27 the authors claim their very indirect estimates are 'direct'. This seems to be a misnomer.*

RESPONSE: We have removed the word 'direct' from line 26-27.

11. *Any comparison of parasites found in wild mosquitoes with those found in human hosts must address the challenge that it is impossible to capture most of the mosquitoes that are biting humans, so the mosquitoes represent a small, (and possibly unrepresentative) sample, so at best the study can investigate statistical correlations.*

RESPONSE: As we note in the Methods, "One morning each week, indoor resting mosquitoes were collected from participant households using vacuum aspiration with Prokopacks," and therefore the collection method of "wild" mosquitos was specifically chosen to capture those most likely to participate in transmission, owing to the endophilic and endophagic preferences of the local vectors (as noted above in #1). We have added to the Discussion: "Similarly, mosquito sampling was necessarily sparser than human sampling, precluding absolute measurement of all transmission events but allowing for relative estimations to onward transmission."

12. *Untreated malaria infections are typically symptomatic for a period early in the infection, followed by a long asymptomatic tail. Because of the lag time in the human host between densities of blood stage infections and infectiousness, (corresponding to the duration of gametocytogenesis) it is to be expected that untreated infections are*

most infectious in the period after a symptomatic attack. It follows that symptomatic and asymptomatic infections, detected at surveys, can be genetically identical with the difference just reflecting the time point at which the sample was drawn. So the authors 'symptomatic' infections were undoubtedly 'asymptomatic' at another (unsampled) time point.

RESPONSE: We detected symptomatic infections in real-time using rapid diagnostic tests and treated these with Artemether-Lumefantrine, therefore preventing these infections from entering an asymptomatic tail phase. We agree that symptomatic infections were almost certainly asymptomatic (or, more specifically, pre-symptomatic) earlier in time, at which point they either were or were not captured on active case detection. We expect that this would bias our results towards the null owing to the classification of this episode solely as "symptomatic" and attributing any transmission to it as a symptomatic infection. We have added to the Discussion: "Conversely, we may have under-detected asymptomatic infections and therefore over-represented symptomatic infections, owing either to the sparse monthly sampling for asymptomatic infections or the inability to capture transmission from symptomatic infections during their asymptomatic/presymptomatic phase. We expect that this would serve mainly to bias our analyses towards the null by providing relatively more opportunities for symptomatic infections to match to mosquitos."

13. The temporal effects that are reported (lines 158-164) could be secondary to any of a number of seasonal epidemiological trends, of which the seasonal pattern of pathology is only one.

RESPONSE: We agree that temporal variation in our transmission setting is governed by many factors. Testing the sources of this variation is beyond the scope of this report on relative transmission from asymptomatic hosts.

14. Before using of genetic differences as a marker of whether the source infection is symptomatic, the investigators need to establish that there are genetic differences between these categories in the markers in the study. However, other studies of Pfcsp and Pfama-1 do not suggest that genotypes at these loci encode virulence factors.

RESPONSE: We agree that these targets do not encode virulence factors. We have clarified in the Results: "Pfcsp and pfama1 were selected owing not to phenotypes associated with their protein products but rather to their sequence diversity, which enables capture of diverse parasite strains and matching strains between hosts."

15. Any relationship of parasite genetics with disease could easily be confounded, for instance by an effect of age. The age of the host is known to affect infectiousness (most likely via parasite densities). It also affects the ratio of asymptomatic/symptomatic periods of infection. The age of the host is also likely to differentially affect the densities

(as the authors recognise, lines 126-128) and hence infectiousness of different parasite genotypes independently of any effect of symptomatic status.

RESPONSE: Agreed. We controlled for age in all the main models, by categorizing into under-5, school-age children (5-15), and adults (> 15y), as described in the Methods in equation 5 and the text “The final model included covariates for village... participant age at study enrollment (categorized: <5 years, 5-15 years, >15 years)...” and presented in the Figure 5 forest plot. The analysis presented in Figure 3 does not include age because, in this sub-analysis, transmission probabilities are compared within individuals, and therefore we do not compare transmission across ages but only within individual people.

16. The different quantities $P(TE_t)$, $P(TE_d)$ and $P(TE_h)$ (see methods and Table 1) do not behave like probabilities (as is illustrated by the need to consider truncation and re-scaling in order to constrain $P(TE_{all})$ to be within the allowable range (equation 4). It is not clear what are the implications of this for the estimates of relative infectiousness of asymptomatic infections, but a model that considered relative rates, rather than probabilities that are not probabilities, would be more convincing.

RESPONSE: The three values we used to calculate a combined probability of transmission, $P(TE_t)$, $P(TE_d)$ and $P(TE_h)$, were calculated to represent relative probabilities of transmission across asymptomatic and symptomatic infections collected in the study and not to be interpreted as absolute probabilities. This interpretation has been clarified in the text (L165-166): “ $P(TE_{aji})$ represented a relative likelihood that a human and mosquito pair that shared parasite haplotypes represented a transmission event and can only be interpreted as an internal ranking.”

17. The ‘pruning’ of incompatible genetic types from the analysis (lines 118-120) also seems likely to introduce bias, though it is not clear in what direction. The fact that a genetic type was not detected does not mean that it was not present.

RESPONSE: We have clarified that there was not any pruning of incompatible genetic types. Instead, we restricted our analysis of mosquito-human pairings to those which met pre-defined criteria on temporal and geographic distance between infections that could plausibly have been transmission events. We have added to the Methods: “... paired each participant’s infection events with all mosquitoes that were collected within 3 kilometers as well as between 7 days prior to and 14 days following the participant infection, in order to constrain the search space for plausible transmission events to within time and distance parameters that are consistent with parasite and mosquito biology.”

18. The assumption of a log-linear relationship (equation 1) needs to be justified. Both

previous empirical studies and theoretical models suggest that the relationship of infectiousness to parasite densities is non-linear. This may be critical for the analysis because parasite densities are a major potential confounder.

RESPONSE: The choice of a log-linear relationship was selected based on a functional form assessment for each continuously-measured covariate: the parasite density, participant age and mosquito abundance (L501 -502). We have added more information to the Supplementary information in a new section "Functional form assessment for continuous variables:" "A functional form assessment was conducted for continuous variables included in the models: parasite density in the participant samples, participant age at study enrollment, and mosquito abundance. The functional form assessment indicated that the optimal coding for parasite density was linear and rescaled to have a mean value of 0.0 due to its interpretability and similar functional form (Table S3). For participant age, the categorical coding (categorized: <5 years, 5-15 years, >15 years) was the best choice, because it had the lowest Akaike information criteria (AIC) value, fit the functional form, and was a commonly used coding of age in malaria literature (Table S4). For mosquito abundance, a binary coding was chosen (expressed as the total number of female *Anopheles* mosquitoes collected within the week following the participant infection stratified at <75 mosquitoes or 75 mosquitoes), because that functional form had the lowest AIC, was easily interpretable, and had a similar functional form to the variable (Table S5)".

19. *The inclusion of both passively and actively detected cases among symptomatic infections may be a source of bias since the time at risk while symptomatic is overrepresented in the dataset.*

RESPONSE: We address this as a limitation but have further clarified this in the Discussion: "Conversely, we may have under-detected asymptomatic infections and therefore over-represented symptomatic infections, owing either to the sparse monthly sampling for asymptomatic infections or the inability to capture transmission from symptomatic infections during their asymptomatic/presymptomatic phase. We expect that this would serve mainly to bias our analyses towards the null by providing relatively more opportunities for symptomatic infections to match to mosquitos."

20. *It is not clear whether the parameter MOI refers to the unique haplotypes at a single time point or among the complete set of samples from the individual.*

RESPONSE: We have clarified this in the Methods: " MOI is the participant's multiplicity of infection (MOI), represented by the number of unique gene haplotypes observed in the participant's infection (I)."

21. *Line 43-44. Reference [1] indicates a gradual decrease in malaria incidence since 2010, not a plateau.*

RESPONSE: Agreed, that was an inapposite geographical metaphor. We have changed this to read: "...prevention efforts, progress in malaria control has stalled since 2010, with 228 million..."

22. *Line 95. The finding that pfmsp multiplicity of infection was higher for mosquitoes than for human hosts is very surprising. In most studies, the majority of infected mosquitoes contain only single oocysts, which should correspond to MOI of two or less (and mosquitoes are unlikely to be superinfected). Do the authors have any explanation for this?*

RESPONSE: We've addressed the issue of MOI in mosquitos in response to #3 above with a new paragraph in the Discussion.

Reviewer #3

23. *Given the longitudinal nature of the dataset, is it possible to estimate the mean duration of an asymptomatic infection? Or even the mean duration of a given haplotype infection within an individual? Given the very quick treatment of symptomatics in this study, it would mean you could look at how the relative contribution might vary in more realistic treatment scenarios. Also given that in reality a lot of symptomatic individuals never seek treatment and just transition to becoming asymptomatic, how does that fit in with this paper? That would suggest advocating for better treatment access rather than MDA approaches I think. I don't want to be the reviewer that suggests you write a whole different paper! But maybe useful to think about what value these bits of information would add*

RESPONSE: We do want to write a different paper! But it will indeed be a different paper: in a follow-on analysis, which we are estimating if the risk of symptomatic disease is associated with the quality, duration, or number of previously-observed haplotypes. Owing to the complexity of both the data structures here and the interpretation of the necessary modeling, these analyses would be pretty far beyond the scope of this paper, in which we've tried to stay focused on the relative contribution of asymptomatic disease to transmission to mosquitos. Given the Reviewer's assistance in the queries below, we would be happy to have her/his review of that paper.

24. *The approach for calculating $P(TE_all)$ makes sense, but I was wondering if you could do a loose validation by looking at counterfactuals, i.e. is there an increased number of shared haplotypes in human-mosquito samples that are <3km compared to*

>3km, is there an increased number of shared haplotypes in samples within your infection window compared to outside it. I think seeing these data would reassure me that you are indeed picking up truly related infections (I'd recommend this as an addition to the supplement, not the main text).

RESPONSE: We compared the number of haplotypes shared in specimens collected less than 3 kilometers of each other to those collected at a distance of 3 kilometers or more and found that there was a lower average number of haplotypes shared between specimens collected at a distance greater than 3 kilometers but the difference was not large. Results are in the supplement (**S9 Fig**).

In addition, we've now included a variety of sensitivity analyses (as noted above for #4) to understand the influences of the constraints on the intervals of distance and time on the model output (**S6 Fig** and **S8 Fig**) as well as an ensemble analysis of *pfmsp* and *pfama1* in a single model (**S10 Fig**). The time and distance models indicate a slight decline in the OR for asymptomatic infections with increasing distance and time, while the ensemble model returns an OR similar to that for the estimate for the *pfmsp*-only model.

We added some text to the methods section describing the distance sensitivity analysis and haplotype sharing comparison: "A sensitivity analysis was conducted to evaluate how changing the distance between specimen collection to allowing specimen collection at a distance greater than 3 kilometers influenced results (**S8 Fig**). We also compared the number of *pfmsp* haplotypes shared within 3 kilometers compared to at a distance of greater than 3 kilometers (**S9 Fig**)."

25. L34-38 : *this sentence isn't very clear – I think the start 'using these haplotypes...' is too unspecific, meaning it's not clear exactly what you're comparing. I'd recommend very briefly describing the method – i.e. 'Using haplotypes to probabilistically link infections in humans and mosquitoes...' or something. And then, 'among individuals that had both symptomatic and asymptomatic infections during the course of the study, we found that asymptomatic infections had double the odds of infecting a mosquito compared to symptomatic infections'.*

RESPONSE: We have changed this sentence to read: "We used these haplotypes to probabilistically link infections in humans and mosquitoes; compared..."

26. L97 : *Interesting that MOI was higher in asymptomatic infections – is this because 1 strain is swamping the reads in symptomatic infections?*

RESPONSE: This is possible. We took a conservative approach to haplotype quality filtering in order to minimize false-discovery, which entailed censoring reads and haplotypes within infections based on two specific criteria that may be influenced by

just such a phenomenon, namely the removal of haplotypes i) supported by less than 3% of the total reads for that infection or ii) which were defined by a 1 SNP difference from a majority haplotype that had > 8X the read abundance. We enforced these fairly strict criteria based on our own empiric validation data (**S1 Fig**) and the reports from other groups that the risk of haplotype false-discovery is enhanced without these criteria. Therefore we don't have enough confidence in the data that could be used to address this issue, namely the quantification within infections of very low-abundance haplotypes.

27. *L97: Are there instances where individuals have asymptomatic infection in the sample prior to their symptomatic episode? Are the haplotypes of the asymptomatic infection the same as the symptomatic haplotype. If an individual had multiple haplotypes in the same before getting sick, it'd be unlikely they'd cleared them all in <1 month.*

RESPONSE: Using all participant infections collected in the study and sequenced for *pfensp*, we observed at least one of the same haplotypes in consecutive infections in 213/841 (25.3%) of infections. Most of the infections with persistent haplotypes were asymptomatic infections that stayed asymptomatic but 37/213 (17.4%) were asymptomatic infections that became symptomatic. In a secondary report, we are more fully investigating the relationship between haplotypes acquired over time in the study participants (as described above in #23), but it was outside of the scope of this study.

28. *L101 : are these 65 individuals a biased sample and therefore unrepresentative? i.e. how does the mean age compare to the whole population, are they a group of individuals that just get bitten more? How does # infections in this group compare to the population as a whole?*

RESPONSE: We have now added a table to the supplement that compares the 65 participants in the within-participant analysis to the 198 participants in the full probabilistic model analysis (**Table S1**):

Table S1. Comparison of participant-mosquito pairs among 65 participants included in within-participant modeling to full data set of all participants

	Analysis data set 65 participants (1565 pairings)	Full data set 198 participants (3727 pairings)	P-value
Participant-level covariates			
Parasite density (parasites/ L), Median (IQR)	290.55 (3654.96)	43.49 (731.76)	<0.001 ^a
Age, N (%)			<0.001 ^b

<5 years	179 (11.44)	438 (11.75)	
5-15 years	1105 (70.61)	1806 (48.46)	
>15 years	281 (17.96)	1483 (39.79)	
Number of pfcsp haplotypes, Median (IQR)	1.00 (2.00)	3.00 (6.00)	0.211 ^a
Number of infections per participant, Median (IQR)	3.00 (2.00)	2.00 (3.00)	<0.001 ^a

Abbreviations: IQR, interquartile range

^aWilcoxon Rank Sum test with continuity correction and Bonferroni correction for repeated measures

^bPearson's χ^2 test with Bonferroni correction for repeated measures

We did observe some differences in the two populations and added text to the results section to reflect this: “This yielded 1565 participant-mosquito pairs for the 225 events; this subset of participants and events was similar to the overall population (Table S1).” We note that the differences between populations are to be expected given the criteria for inclusion in this subset, namely that the requirement that a subset person have at least 1 asymptomatic and 1 symptomatic infection resulted in the median number of infections per participant being slightly higher (3) than the overall population (2), and the requirement to have at least 1 symptomatic infection (which have generally higher densities) resulted in densities being a bit higher.

29.L108 : *in the logistic regression, did you account for the fact that there were multiple timepoints from the same individual? Did you account for the frequency of the haplotype in the population? Did you transform the parasite density data (i.e. log10) as otherwise models can get really skewed by very high numbers.*

RESPONSE: The logistic regression models included a random intercept at the participant level to take into account that there were multiple timepoints from the same individual (L497).

We accounted for the frequency of the haplotypes in the population when creating the $P(TE_h)$ term of the probabilistic model by calculating the prevalence of the haplotype in the study population and upweighting the sharing of rarer haplotypes to reflect a higher probability that the sharing of a rarer haplotype represented a true transmission event.

Parasite density was rescaled and centered at a mean of 0 (L455-479). Text describing this has been added to the Methods: (L506-507): “To reduce skew for the multi-level model, parasite density was centered and rescaled to have a mean of 0.”

30.L159-160: *normally ‘prevalence of asymptomatic infections’ refers to proportion of*

whole population that have an asymptomatic infection. I think you mean something like ‘monthly proportion of all infections that are asymptomatic’.

RESPONSE: We changed this as suggested to “...as a function of the monthly proportion of all infections that were asymptomatic, which varied...”

31. L242 : *I disagree that this access is generalizable – from the last world malaria report “Based on 19 household surveys conducted in sub-Saharan Africa between 2015 and 2017, the percentage of children aged under 5 years with a fever who received any antimalarial drug was 29% (IQR: 15–48%).” – and given that symptomatic kids are more likely to get treatment than symptomatic adults, I think the population-level figure will be even lower than this 29% - a long way away from the likely ~90% in this study. Also here, referring back to my first comment in major revisions – about how in reality some symptomatic individuals never seek treatment and just become asymptomatic over time.*

RESPONSE: Agreed, we were overly optimistic when discussing this limitation. We have changed the Discussion to read: “Symptomatic infections were quickly diagnosed and treated with effective therapy under our protocol which likely reduced the duration of these infections and therefore limited their transmission potential. This access to diagnosis and treatment is higher than is generally available across sub-Saharan Africa, though recent reports indicate gradual improvement in quality clinical management.”

32. L 357 – *why 14 days after – if the individual because asexual parasite positive on the day of screening, they might not even develop gametocytes for 10 days or so, so I think 14 days might be too stringent – did you look at any sensitivity around this value?*

RESPONSE: We’ve now added a sensitivity analysis in which we’ve computed the multilevel logistic regression model in Equation 5 using different time windows for what could be considered a human-to-mosquito malaria transmission event. The time windows allowed mosquitoes to be collected from 14 days after the participant infection up to 30 days after. Results are in the Supplement (**S6 Fig**) with the following caption: “S6 Fig. Sensitivity analysis for probability of transmission over time. A sensitivity analysis was performed to understand the influence of the time window allotted for mosquito matching to human infections (y-axis) on the outcome of the relative likelihood of transmission from an asymptomatic infection. To do so, the multi-level logistic regression model comparing probabilities of transmission events P(TE) between asymptomatic and symptomatic infections was computed using time windows for mosquito collection in which the window consistently began 7 days prior to the human infection (+7) but the interval after the human infection was allowed to vary between 30 and 14 days (i.e. -30 to -14 days). The *pfmsp* haplotypes were used for this sensitivity analysis.” This text was added to the methods: “A

sensitivity analysis was conducted to assess how differences in the time window chosen affected results, expanding the time window to allow mosquitoes to be collected up to 30 days after the participant's infection (S6 Fig)."

33.L402 – by rescaling the probabilities, aren't you in effect just assign arbitrary weights to each factor? You mention you do a sensitivity analyses but I couldn't find the details? I'd recommend logging the equation and then assigning weights to each component and varying the beta values here to see if that varies the results much i.e. $\log(P(\text{TE}_{\text{all}})) = \beta_1 * \log(P(\text{TE}_{\text{t}})) + \beta_2 * \log(P(\text{TE}_{\text{d}})) + \beta_3 * \log(P(\text{TE}_{\text{h}}))$

RESPONSE: We have now added sensitivity analyses to better understand if varying the time, distance, and haplotype variables and affect the relationship between asymptomatic versus asymptomatic malaria and the probability of transmission to mosquitoes using Equation 5. Results are in the supplement (S6, S8, and S10 Fig).

34.L81/Fig 1 : I was initially confused about the fact there were 239 participants but the bars in fig 1, row 3 only had bars 60 high, but figured it out – maybe useful to add a note on the scheduling of ACD visits in the villages to clarify this?

RESPONSE: We've added to the Figure 1 caption some operational detail: "These monthly visits were conducted in different weeks for each of the 3 villages, with additional re-visits if needed to sample enrolled participants who were absent for the initial visit."

35.L89 : can you be explicit about why the majority of results for pfama1 are in the supplement? Is this known to be a less good gene for this type of analyses? Or were results just consistently less significant?

RESPONSE: To limit the amount of results and figures included in the main text, we chose to put results for one gene target in the supplement. We highlighted the *pfmsp* results in the main text of the manuscript and had *pfama1* results in the supplement owing partially to the consistency of the analyses as well as the ability to better validate the quality of the *pfmsp* haplotypes by comparing to the deposited *pfmsp* haplotypes contributed by Neafsey *et al.* *NEJM* 2015 (presented in S2 Fig), which served as a unique external validation of many of the variant positions observed in *pfmsp*.

36.L130 : It would be interesting to see the full range of values as well as the IQR here

RESPONSE: The text has been updated to show the full range of values: "Across the all pairings, the median number of haplotypes shared within a participant-

mosquito pair was 1 (range: 0 to 8, IQR: 0 to 2) for asymptomatic and 0 (range: 0 to 7, IQR: 0 to 1) for symptomatic infections.”

37.L179 : ‘consonant’ should be ‘consistent’

RESPONSE: Changed to “consistent.”

38.L205 – not sure either of the papers referenced here really look at ‘predictive’ models. There is another Okell paper looking at modelling MSAT/MDA approaches – do you mean this one <https://journals.plos.org/plosone/article?id=10.1371/journal.pone.0020179>

RESPONSE: We updated the language here to better describe the papers: “Other studies have used modelling approaches to estimate how transmission dynamics could change in a more realistic setting, finding that submicroscopic infections are a large source of malaria spread [34,35].”

39.L274 – I think useful to highlight that PCR positive asymptomatic individuals were not treated

RESPONSE: We’ve added to the Methods a bit later: “The DBS were tested for *P. falciparum* parasites using real-time PCR post-hoc (see below), and therefore parasites detected in asymptomatic people were not treated.”

40.L348 – clearer to just say ‘multiplied’ rather than ‘aggregated’

RESPONSE: Changed to “multiplied.”

41.L370 – clearer to just say the number? ‘For example, by 0.66km from the human host, the probability is already low (14%).’

RESPONSE: We clarified the probabilities in the sentence to now say: “For example, by 0.66 kilometers from the participant, which was the maximum distance blood fed *Anopheles* mosquitoes were observed to fly in a Kilifi study [73], the probability of transmission was already low (14%) and at 3 kilometers it had dropped to 0% entirely.”

42.L409 – define i and j

RESPONSE: Variables i and j in the model are now clearly defined in the text: “We included a random intercept at the participant level () to account for repeated measures for participants who experienced multiple malaria infections (asymptomatic or symptomatic) throughout the study. To consider different

transmission intensities between households, we included a random intercept at the household level ().”

43. L411 – do the village dummy variables need i and j indexes?

RESPONSE: The i and j indexes have been removed from the village dummy variables.

44. L441 – define a and t in the definition of the contribution to the infectious reservoir

RESPONSE: The language has been clarified to indicate that a represents asymptomatic infections, s symptomatic infections, and t time represented by month.

45. L445 – is this proportion of all infections that were symptomatic? A bit unclear

RESPONSE: We have clarified and aligned with the text edit for #30 above by changing to: “1. P_{at} represents the proportion of all infections that were asymptomatic during each month (t) of follow-up” and “2. P_{st} represents the proportion of all infections that were symptomatic during each month (t) of follow-up.”

46. Fig 1 – add in prevalence lines on a second y-axis for all three graphs?

RESPONSE: Due to the difference in sampling scheme between the sample types (monthly for asymptomatic infections, as needed for symptomatic infections, and weekly for mosquito infections), we chose to represent the number of samples collected as bars. Prevalence lines for each week would not accurately represent the samples collected because of the differences in sampling schemes. To make interpretation easier for figure 1, we used stacked bars to compare the number of samples that were *P. falciparum* positive to the total number of samples collected and used a standardized y-axis of 100 samples for all three sample types (asymptomatic infection, symptomatic infection, mosquito abdomen).

47. Fig 2 – hard to see the small bars – add $n=XX$ for each bar in left and right margins of graph?

RESPONSE: To avoid visual clutter in this Figure 2 and Fig S12, we’ve added a table in the supplementary material to clearly represent haplotype counts across all three sample types. We added text regarding this change to the main text (L870871): “A full plot of all 229 *pfmsp* haplotypes and a table of the counts for these haplotypes across sample types is in the supplement (**S12 Fig, Table S6**).”

48. Fig 4D – would this be easier to interpret as a histogram?

RESPONSE: Figure 4D has been updated to now be a histogram instead of a density plot.

Thanks for your consideration of our manuscript for *Nature Communications* and please contact us for any further clarifications or additions.

Reviewers' Comments:

Reviewer #1:

Remarks to the Author:

Revised manuscript: Genotyping of cognate *Plasmodium falciparum* parasites in naturally-infected human and mosquito hosts to estimate the contribution of asymptomatic infection to onward malaria transmission.

The authors have addressed my comments, and the comments of the other reviewers seriously, and with careful thought and detail. Thank You! I have a few minor clarifications based on these responses which I believe to be very minor in nature.

Reviewer #1 (me)

Response to comment 3 – the statement "...the diploid nature of oocyst forms would be expected to increase diversity compared with the haploid asexual forms..." I disagree with the haploid forms which give rise to the diploids contain the same level of genetic diversity as the haploids themselves. That is, the genetic makeup of a diploid is sampled from the genetic diversity of haploid gametes taken up in a blood meal. I don't see how diploids increase genetic diversity, especially at individual loci.

Response to comment 5 – I appreciate the additional detail on selection acting on these markers. Though perhaps my original comment was not precise, I would expect that immune recognition may impact the presence or absence of specific alleles of *csp* or *ama1*. This may influence the detection of the actual MOI in the infection as allele specific immunity erases or represses certain haplotypes. I hoped that the authors would include a brief discussion on how MOI estimates may differ at loci under immune selection such as these. However, I agree that the exceptional diversity at these markers is a compelling case for their use.

Ian Cheeseman

Reviewer #2:

Remarks to the Author:

The authors have responded to the issues raised in the initial review but I am not convinced by several of their responses:

1. The fundamental point remains that their claim for originality of the principal result remains overblown in the revised manuscript. The authors' approach is complementary to other ways of studying the contribution to malaria transmission of different population groups but provides much less direct data than studies that track transmission at the level of the individual, rather than the group.

It has been known since the 1950s that most malaria transmission in high transmission areas arises from asymptomatic individuals. The earliest studies of infectiousness used direct feeding of mosquitoes (Muirhead-Thomson, 1954, 1957). From a methodological point of view this provides much more solid and direct evidence than the authors' genetic analysis, though such studies would not be given ethical approval nowadays. The authors could argue that Muirhead-Thomson did not concern himself with whether the participants of his study had acute illness at the time of sampling, but it must have been obvious that most of them didn't. Most people in very high transmission settings are asymptomatic at any one time and this has been no surprise since the phenomenon was first described by Robert Koch in 1900.

The strongest data on the relationship between symptoms and infectiousness in *P. falciparum* malaria is that of artificial infections used to treat neurosyphilis. Hundreds of time-series provide data on the timing of acute illness and infectiousness, showing how these show different temporal dynamics and

hence provide an explanation for why most of the infectious reservoir is asymptomatic. The authors of the current manuscript have implicitly excluded this evidence by qualifying their claims as referring to natural infections.

The authors do refer to the evidence from more recent membrane feeding studies, which have added to the evidence base more recently, essentially supporting the conclusions from earlier direct feeding and experimental/therapeutic studies. Estimates of the contribution of asymptomatic infections to transmission from membrane feeding studies are surely more robust than those from this study.

2. The surprising and novel result in the paper is the finding of much higher multiplicity at the *csp* locus in mosquitoes than in humans. The opposite would be expected because mosquito blood meals sample only a very small blood volume. The authors' suggestion that "This could have resulted from the transmission to mosquitoes of cryptic haplotypes that were undetectable in asexual human infections" is not plausible without further explanation since there is no obvious reason why infections should more often be cryptic in the human host than in the mosquito. As the authors observe, the number of oocysts in a mosquito is typically small; each oocyst can at most contain 4 alleles. The authors explain this by referring to evidence that *Anopheles gambiae* often takes multiple blood meals in the same oviposition cycle, but it is not clear that this can explain the excess, especially since taking multiple blood meals is not considered the norm except perhaps during the first oviposition cycle. It would be useful to see some mathematical elaboration of how the authors suggest this result arises.

3. In addition, I contend that the authors response to Reviewer 1's first point: 'What is the chance of capturing ... ', does not really answer the question. If I understand the methodology correctly, then the probability that an event is captured arises as a normalising constant which is not estimated. This probability is presumably very low, as the proportion of mosquito population that is captured and analysed is tiny.

Muirhead-Thomson, AJTMH 6, (1957), p. 971 – 979.

Muirhead-Thomson, Trans R Soc Trop Med Hyg. 1954 May;48(3):208-25. doi: 10.1016/0035-9203(54)90067-x.

Reviewer #3:

Remarks to the Author:

The authors have responded to all my points well and I think the manuscript is ready for publication.

Reviewer # 1

1. Response to comment 3 – the statement “...the diploid nature of oocyst forms would be expected to increase diversity compared with the haploid asexual forms...” I disagree with the haploid forms which give rise to the diploids contain the same level of genetic diversity as the haploids themselves. That is, the genetic makeup of a diploid is sampled from the genetic diversity of haploid gametes taken up in a blood meal. I don't see how diploids increase genetic diversity, especially at individual loci.

RESPONSE: Agreed, and this was poorly stated. We have removed this sentence (and replaced with that in #2 below).

2. *Response to comment 5 – I appreciate the additional detail on selection acting on these markers. Though perhaps my original comment was not precise, I would expect that immune recognition may impact the presence or absence of specific alleles of csp or ama1. This may influence the detection of the actual MOI in the infection as allele specific immunity erases or represses certain haplotypes. I hoped that the authors would include a brief discussion on how MOI estimates may differ at loci under immune selection such as these. However, I agree that the exceptional diversity at these markers is a compelling case for their use.*

RESPONSE: Right, now we are picking up what Review #1 is putting down. We've added to the Discussion this point: "On a related note, partial immune recognition of expressed circumsporozoite protein or apical membrane antigen-1 variants, which are expressed in the liver or blood stage respectively, may have served to differentially limit the densities of certain variants below the limits of detection in human infections while allowing passage to and propagation in mosquitos."

Reviewer #2

3. *The fundamental point remains that their claim for originality of the principal result remains overblown in the revised manuscript. The authors' approach is complementary to other ways of studying the contribution to malaria transmission of different population groups but provides much less direct data than studies that track transmission at the level of the individual, rather than the group.*

It has been known since the 1950s that most malaria transmission in high transmission areas arises from asymptomatic individuals. The earliest studies of infectiousness used direct feeding of mosquitoes (Muirhead-Thomson, 1954, 1957). From a methodological point of view this provides much more solid and direct evidence than the authors' genetic analysis, though such studies would not be given ethical approval nowadays. The authors could argue that Muirhead-Thomson did not concern himself with whether the participants of his study had acute illness at the time of sampling, but it must have been obvious that most of them didn't. Most people in very high transmission settings are asymptomatic at any one time and this has been no surprise since the phenomenon was first described by Robert Koch in 1900.

RESPONSE: We agree that Muirhead-Thomson did not report on the presence or absence of symptoms, and therefore we can't speculate as to whether the transmissions that he observed in his feedings upon infected individuals were from symptomatic or from asymptomatic people. What Muirhead-Thomson does make clear is that, on a community scale, transmissibility is a function of multiple factors that are beyond those captured in his experimental feeds. He specifically alludes to the differential biting by Anophelines of people of different ages and the age-structure of the local community, and to this we would add the following factors that influence the transmissibility of patent blood-stage infections that are not measurable by experimental feeds: the age structure of biting *Anopheles gambiae*, the biting

behaviors of wild-caught (as opposed to laboratory-reared) *A. gambiae*, the use of insecticide-treated bednets by participants, mediation of biting attractiveness to Anophelines by patent blood-stage infections or other cryptic factors, the durations of types of *P. falciparum* infections, variations in gametocyte density during infection, and human behavioral changes aside from bednet use that alter the availability to Anophelines. Although each of these factors could be estimated crudely and transmissibility could be modeled as the product of these on individual and community scales, we opted for our described approach. We've tried to draw a distinction to clarify how the insights afforded by our study complement those prior to ours in the Introduction: "Such controlled feeding studies are critical to understand the fundamental biology of parasite transmission, and studies in natural, uncontrolled settings are necessary to confidently extend these insights to understand how they shape disease epidemiology."

We reference these prior studies and their limitations in the Introduction and in the Discussion: "Mosquito feeding experiments employing either direct skin or membrane feeding fail to represent numerous participant-, mosquito-, and parasite-related factors that are critical to transmission. These critical factors include variance among mosquito vectors in biting preferences [28], behaviors [26], and success [29]; among parasites in replication rates [27] and gametocyte production [30]; and among participants in exposure to vectors [31] and care-seeking behavior [32]. Similarly, this complexity also confounds the use of gametocyte prevalence or density as a proxy for transmission [33], which may more precisely define which infections can rather than do transmit."

4. The strongest data on the relationship between symptoms and infectiousness in P. falciparum malaria is that of artificial infections used to treat neurosyphilis. Hundreds of time-series provide data on the timing of acute illness and infectiousness, showing how these show different temporal dynamics and hence provide an explanation for why most of the infectious reservoir is asymptomatic. The authors of the current manuscript have implicitly excluded this evidence by qualifying their claims as referring to natural infections.

The authors do refer to the evidence from more recent membrane feeding studies, which have added to the evidence base more recently, essentially supporting the conclusions from earlier direct feeding and experimental/therapeutic studies. Estimates of the contribution of asymptomatic infections to transmission from membrane feeding studies are surely more robust than those from this study.

RESPONSE: As described in the Introduction and the Discussion, there are myriad prior studies that investigate this important question, and each of these have important limitations, as described in #3 above. The data from neurosyphilis studies are subject to these same limitations which undermine the ability to confidently project their findings onto natural settings, in that they are measured from selected individual patients in highly controlled settings and do not capture the variables cited above.

4. *The surprising and novel result in the paper is the finding of much higher multiplicity at the csp locus in mosquitoes than in humans. The opposite would be expected because mosquito blood meals sample only a very small blood volume. The authors' suggestion that "This could have resulted from the transmission to mosquitos of cryptic haplotypes that were undetectable in asexual human infections" is not plausible without further explanation since there is no obvious reason why infections should more often be cryptic in the human host than in the mosquito.*

RESPONSE: The detection of genotypes in a mosquito that were not found in the blood upon which they fed has been reported for both *P. falciparum* and *P. vivax* infections, and we have added this to the Discussion: "...that were undetectable in asexual human infections, as has been reported with both *P. falciparum* and *P. vivax*, [Balasubramanian S et al. *J Infect Dis* 2020] although both sample types were processed..."

5. *As the authors observe, the number of oocysts in a mosquito is typically small; each oocyst can at most contain 4 alleles. The authors explain this by referring to evidence that *Anopheles gambiae* often takes multiple blood meals in the same oviposition cycle, but it is not clear that this can explain the excess, especially since taking multiple blood meals is not considered the norm except perhaps during the first oviposition cycle. It would be useful to see some mathematical elaboration of how the authors suggest this result arises.*

RESPONSE: We agree that traditional dogma states that Anopheline mosquitos take only a single bloodmeal during each ovarian cycle, but recent data support the notion that a wide variety of species – like other mosquito genera -- take more than one meal. *An. gambiae* specifically takes 2 bloodmeals during the first cycle, and a recent study in Burkina Faso reported that 151/794 (19%) of wild-caught Anophelines (principally *A. gambiae*) with typable human DNA in their abdomen had fed on more than one human [Guelbeogo WM et al. *eLife* 2018; 7: e32625] We have added detail to this sentence to support this notion: "Finally, given evidence that *Anopheles gambiae* can take multiple bloodmeals per gonotrophic cycle [Gillies MT *Annals of Tropical Medicine and Parasitology* 1954; Scott TW *Am J Trop Med Hyg* 2006] [44], and that this behavior may be enhanced by an existing sporozoite infection of the mosquito, [Koella JC *Proc Biol Sci* 1998] these oocysts may represent an accumulation..."

6. *In addition, I contend that the authors response to Reviewer 1's first point: 'What is the chance of capturing', 'does not really answer the question. If I understand the methodology correctly, then the probability that an event is captured arises as a normalising constant which is not estimated. This probability is presumably very low, as the proportion of mosquito population that is captured and analysed is tiny.*

RESPONSE: We agree that the absolute likelihood of capturing a transmission event is likely low, given the limits on what can be observed in a natural setting. Therefore, rather than absolute estimates of event rates, the focus of this report is

on the relative events between asymptomatic and symptomatic people. The infeasibility of capturing all fed mosquitos is acknowledged as a limitation: “Similarly, mosquito sampling was necessarily sparser than human sampling, precluding absolute measurement of all transmission events but allowing for relative estimations to onward transmission.”

Reviewer #3

None

Thanks for your consideration of our manuscript for *Nature Communications* and please contact us for any further clarifications or additions.

Reviewers' Comments:

Reviewer #2:

Remarks to the Author:

It beggars belief that the authors have sampled a large proportion of the parasite population in either humans, or (especially) mosquitoes, so in principle the probabilistic linkage should only work if distinct genotypes are associated with symptomatic infections. But the loci they have typed are not reproducibly associated with variation in virulence, and any one infection is asymptomatic for part of the time and symptomatic at other times.

There is no consideration of alternative explanations for the most surprising results (e.g. can they discount lab artefacts): the finding of much higher multiplicity at the csp locus in mosquitoes than in humans runs counter to the findings of other studies that find higher multiplicity in humans. The cited literature is arguing that some mosquitoes take two feeds, not that on average they take half-a-dozen, which is what would be required to explain this in terms of multiple feeds per cycle.

Reviewer

1. *It beggars belief that the authors have sampled a large proportion of the parasite population in either humans, or (especially) mosquitoes, so in principle the probabilistic linkage should only work if distinct genotypes are associated with symptomatic infections. But the loci they have typed are not reproducibly associated with variation in virulence, and any one infection is asymptomatic for part of the time and symptomatic at other times.*

RESPONSE: An association between markers and symptomatology/virulence is not a necessary premise of this study, in which we merely used the genetic marks as tracers by which to link an infected person with an infected mosquito, and then “weighted” these linked dyads by factors that mediated the credibility of that link truly representing a transmission event (distance, time interval, number and quality of shared links).

2. *There is no consideration of alternative explanations for the most surprising results (e.g. can they discount lab artefacts): the finding of much higher multiplicity at the csp locus in mosquitoes than in humans runs counter to the findings of other studies that find higher multiplicity in humans. The cited literature is arguing that some mosquitoes take two feeds, not that on average they take half-a-dozen, which is what would be required to explain this in terms of multiple feeds per cycle.*

RESPONSE: Re: the possibility of lab artifacts, as included in the penultimate Discussion paragraph: “Finally, many infections in participants and mosquitoes had low parasite densities, which increases the risk of haplotype false discovery [58]. To mitigate this risk, we enforced stringent haplotype censoring based on read quality and haplotype abundance consistent with prior studies [58–60].”

Re: other studies that report parasite genetic diversity in cognate mosquito and humans hosts, we cannot identify the specific study to which the reviewer is alluding. In one study (Mendes C *et al. Malaria J* 2013; 12: 114) analyzing samples collected

in a highly-endemic setting in Equatorial Guinea not dissimilar to ours, the multiplicity of infection as assessed by PCR-RFLP of a size-polymorphic gene was similar in humans (2.11) and mosquitos (1.91). Our results do contrast with this, which is likely a result of our use of a genotyping approach (amplicon deep sequencing) that is more sensitive to minority variants, as noted in the Discussion: "... amplicon deep sequencing enabled this approach with its technical ability to capture minority variants within mixed infections [40]...."

As reported herein, the median multiplicity of infection (using *pfmsp*) was 3 in asymptomatic people and 6 in mosquitoes, meaning that, on average, a mosquito would need only feed upon 2 people with asymptomatic infections to yield the median MOI estimate.

Thanks for your consideration of our manuscript for *Nature Communications* and please contact us for any further clarifications or additions.